# Renovating Names in Open-Vocabulary Segmentation Benchmarks

**Haiwen Huang**[1,2,3]    **Songyou Peng**[3,4,5]    **Dan Zhang**[6]    **Andreas Geiger**[2,3]

[1] Bosch IoC Lab, University of Tübingen    [3] Tübingen AI Center
[3] Autonomous Vision Group, University of Tübingen    [4] ETH Zurich
[5] MPI for Intelligent Systems, Tübingen    [6] Bosch Center for Artificial Intelligence

## Abstract

Names are essential to both human cognition and vision-language models. Open-vocabulary models utilize class names as text prompts to generalize to categories unseen during training. However, the precision of these names is often overlooked in existing datasets. In this paper, we address this underexplored problem by presenting a framework for "renovating" names in open-vocabulary segmentation benchmarks (RENOVATE). Our framework features a renaming model that enhances the quality of names for each visual segment. Through experiments, we demonstrate that our renovated names help train stronger open-vocabulary models with up to 15% relative improvement and significantly enhance training efficiency with improved data quality. We also show that our renovated names improve evaluation by better measuring misclassification and enabling fine-grained model analysis. We provide our code and relabelings for several popular segmentation datasets to the research community on our project page: https://andrehuang.github.io/renovate/ .

## 1 Introduction

> Categorizations which humans make of the concrete world are not arbitrary but highly determined. In taxonomies of concrete objects, there is one level of abstraction at which the most basic category cuts are made.
>
> — Eleanor Rosch, *Basic Objects in Natural Categories* [1]

We use names every day. Imagine wandering the trails of a national park – do you pause by a "lake" or simply a body of "water"? While driving through urban sprawl, do you see "trees" or "vegetation" along the road? Our instinctual use of terms like "lake" and "trees" illustrates the human propensity to categorize the world around us into "basic categories" — rich in information and readily identifiable without unnecessary complexity [1, 2, 3]. Such categorization is fundamental to human cognition and communication.

In stark contrast, recent advancements in open-vocabulary models [4, 5, 6, 7, 8, 9, 10, 11, 12] struggle to replicate this nuanced aspect of human categorization. While these models have made strides in generalizing to both familiar and novel categories via textual prompts, they are often hampered by the imprecise and sometimes wrong names provided in existing benchmarks. In fact, most datasets are labeled with class names that serve merely as identifiers to distinguish classes within a dataset, rather than descriptive labels aligning with the "basic categories" that match with the actual visual contents. As shown in Fig. 1, existing names are often inaccurate, too general, or lack enough context, leading to discrepancies between the model's outputs and the actual visual segments. This misalignment indicates a pressing need for a reassessment and refinement of name labeling prac-

tices. By adopting a naming scheme that aligns more closely with human categorization, we can pave the way for enhanced open-vocabulary generalization and more accurate model evaluation.

In this work, we focus on renovating the names for open-vocabulary segmentation benchmarks, as their dense annotations pose a greater challenge and offer broader applicability than other recognition tasks. The importance of names in open-vocabulary segmentation is often overlooked, with one exception, OpenSeg [5], which manually inspected and modified the class names of several segmentation benchmarks, resulting in considerable performance gains of their proposed model. While their work demonstrated the importance of choosing good names, a manual approach is subjective and difficult to scale. To date, there exists no systematic study on how to rename benchmarks in a scalable and principled way. Our work is the first attempt to tackle this challenge.

In this work, we introduce a scalable, simple, yet general method for renaming segmentation benchmarks that outperforms manual label efforts like OpenSeg [4]. Our approach leverages foundation models to automate the renaming process, significantly reducing the manual labor traditionally required. Towards this goal, our method can be divided into two steps. First, we narrow down the name search space from the whole language space to a curated list of candidate names. This list is generated by leveraging the original class names with contextually relevant nouns extracted from visual contents via an image captioning model, thereby streamlining the search for names. Next, we employ a specially trained renaming model to identify and select the best-matching candidate name for every ground-truth segmentation mask. In this way, we match an individual name for each instance without any extra human annotations.

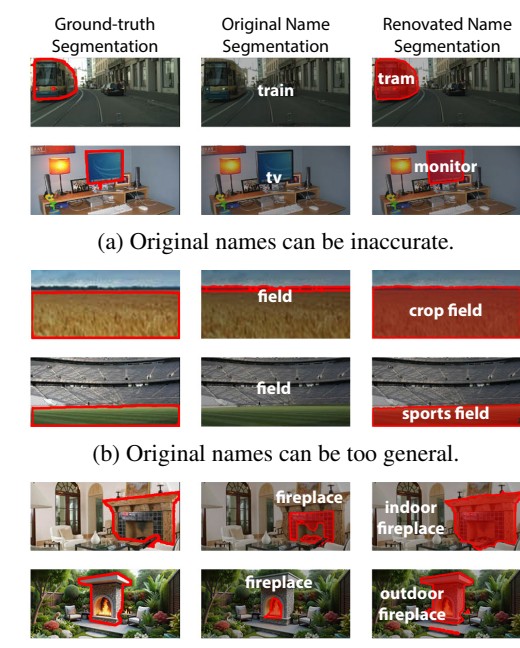

(a) Original names can be inaccurate.

(b) Original names can be too general.

(c) Original names can lack contexts.

Figure 1: **Problems of names in current segmentation benchmarks.** We demonstrate examples from well-known datasets: MS COCO [13], ADE20K [14], and Cityscapes [15]. Our renovated names are visually more aligned and help models to generalize better.

To demonstrate the value of RENOVATE names, we explore two practical applications. We first use our renovated names for training open-vocabulary models and show that they substantially enhance generalization performance across various benchmarks by up to 15%, indicating a more accurate alignment with visual segments. Our new names also significantly increase training efficiency thanks to their enriched textual information. Second, we apply our renovated names to improve evaluation of open-vocabulary segmentation. Specifically, we reveal that current pre-trained models are making many "benign" misclassifications, *i.e.*, misclassified names that are semantically close to the ground-truth names. RENOVATE names enable more fine-grained model performance analysis, providing valuable information for further model enhancement and dataset curation.

In summary, our work makes the following contributions:

- We point out the naming problem in existing open-vocabulary segmentation benchmarks.

- We propose a scalable, simple, and general framework for automatic dataset renaming.

- We demonstrate that our RENOVATE names help to train models with stronger open-vocabulary capabilities and refine evaluation benchmarks, providing valuable insights for future research.

## 2   Related Work

**Open-Vocabulary Segmentation Methods.** Prior work in open-vocabulary segmentation [5, 6, 7, 8, 16, 17, 18, 19, 20] focused on adapting vision-language models (VLMs) [21, 22, 23] to the seg-

| Original name | Context names | Candidate names |
|---|---|---|
| field | lush, field, sky, green, grass, tree, road, hillside, grassy, rural | rural field, roadside field, green field, crop field, sports field, grassland, grassy hillside |
| box | room, stool, chair, floor, market, table, food, lamp, paper | storage box, packing box, file box, box container decorative box, display box, paper box, food box |
| cradle | room, infant bed, nursery, dresser, carpet, bedroom, chair, bureau, lamp, armchair | bedroom cradle, cradle, infant bed, nursery cradle, baby cradle, infant cradle, wooden cradle |

Table 1: **Examples of context names and generated candidate names** for three selected classes from ADE20K. Context names are key to comprehending general terms such as "field" and "box" and disambiguating polysemous terms like "cradle", which, in this context, refers to a baby bed rather than a phone cradle or a mining tool.

mentation task without compromising pre-trained vision-langauge alignment. These efforts have concentrated mainly on the architecture design, exploring different backbones [24, 25], mask proposal branch designs [19, 5], and mask feature extraction techniques [7, 26, 27]. Our work differs from them in that we aim to adapt pre-trained vision-language representations for segment-wise name matching through our novel designs like candidate name generation techniques and our specialized renaming model.

**Names in Open-Vocabulary Segmentation Benchmarks.** Despite its apparent importance, the naming problem in open-vocabulary segmentation benchmarks has not received adequate attention. An exception is the work by [28], which proposes to learn class-specific word embeddings for improved model adaptation to new datasets. However, these learned embeddings are not in the language space and cannot be used by other models. In addition, some works [29, 30] propose to decompose the class names into a set of attributes or prototypes for open-vocabulary segmentation. To the best of our knowledge, OpenSeg [5] is the only work that proposes to directly improve the quality of the names. However, their approach relies on manual inspection of the datasets, which can be subjective and hard to scale. In this paper, we systematically explore the renaming challenge, culminating in a scalable, simple, yet general framework to overcome these limitations.

## 3 RENOVATE: Renaming Segmentation Benchmarks

Our method RENOVATE aims to rename the original names in a given dataset with ground-truth mask annotations. As shown in Fig. 2, we first generate a pool of candidate names, covering potential variants of the original class names. Then, we train a specially designed renaming model that can measure the alignment between segments in pixel space and names in language space. Equipped with the renaming model, we select the top-matching names from the candidate pool for each segment, thereby producing segment-level, refined names that offer enhanced accuracy and details in characterizing the dataset, see Fig. 3 and Fig. 4.

### 3.1 Generating candidate names

We use GPT-4 [31] for creating a pool of candidate names. A naive solution is to prompt GPT-4 with the original name and ask it to generate synonyms and hierarchical concepts. However, since the original names are often too general and ambiguous, GPT-4 does not have sufficient knowledge to generate high-quality candidates. Therefore, we propose to exploit the visual contents for generating some "context names" (Table 1) that assist GPT-4 in comprehending the category's meaning prior to generating candidate names.

As shown in Fig. 2, for each category, we use an image captioning method to process all training images that contain that specific category based on ground-truth annotations. From the generated captions, we further extract nouns by text parsing and filtering as done in CaSED [32]. We observe that nouns appearing more frequently tend to offer deeper insights into the typical environments or traits associated with the category. Therefore, we construct *context names* by selecting the top 10 most recurrent nouns for each category. We then use them as additional inputs alongside the original class names to prompt GPT-4.

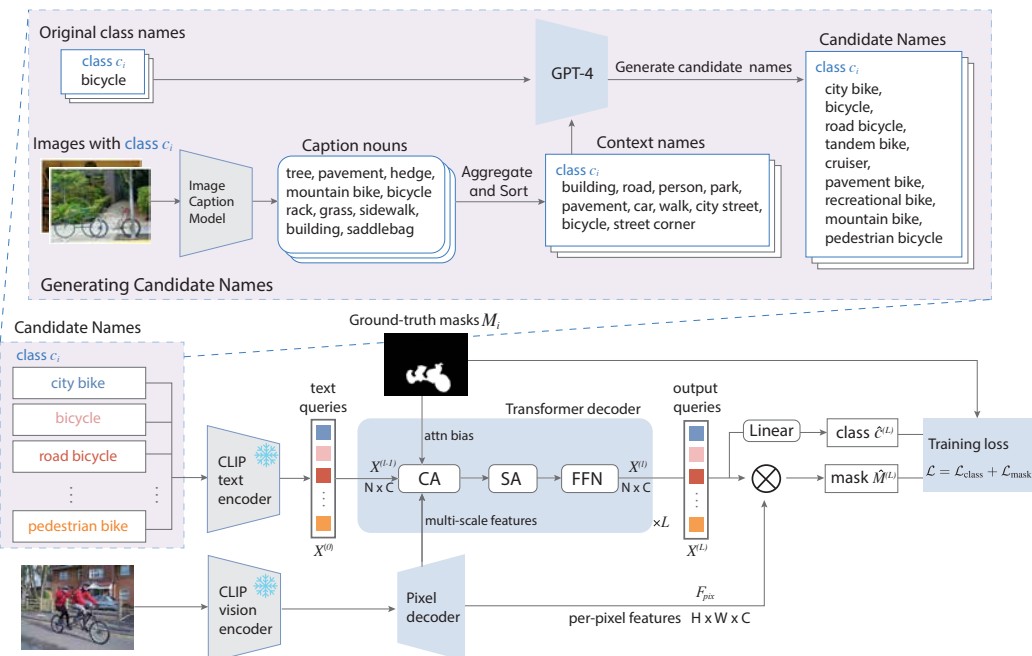

Figure 2: **Overview of candidate name generation and renaming model training.** We generate candidate names based on the context names and train the renaming model to match them with the segments. For illustration clarity, we show only one segment. In practice, multiple segments are jointly trained, pairing with the text queries.

Specifically, we instruct GPT-4 to generate two types of candidate names: (1) *Synonyms or subcategories*, which can expand the lexical scope of the overly general class name. For instance, the class name "grass" could yield variations like "lawn" or "turf". For this type, synonyms and subcategories are considered together, acknowledging their often interchangeable nature. (2) *Providing short, distinct contexts* to polysemous names. As an example, the ambiguous name "fan" could be elaborated into "ceiling fan" or "floor fan", offering clearer specificity. As shown in Table 1, the 5-10 candidate names generated by GPT-4 are more precise than the original names, reflecting the context information encoded by context names. We provide detailed GPT-4 prompts in the supplement.

## 3.2 Training for candidate name selection

Among the generated candidates, we will only keep those that are better aligned with the visual contents. Particularly, we train a model that can assess the alignment between each name and segment, using segmentation as the proxy task. Conceptually, a candidate name that well describes the segment in the image should help the model recover its ground-truth mask and make a correct classification. Therefore, the model should allow vision-language interaction, and the supervision signal should encourage to use textual information for mask prediction and classification.

Our renaming model is illustrated in Fig. 2. Its vision part starts from a CLIP-based vision backbone followed by a pixel decoder that gradually upscales the backbone features to generate per-pixel embeddings $F_{\text{pix}} \in \mathbb{R}^{H \times W \times C}$. The candidate names of the categories present in the image are processed by the CLIP text encoder. The interaction of textual and visual information takes place at the transformer decoder. The initial queries $X^{(0)}$ are from the text embeddings, and then updated by the transformer decoder consisting of $L$ blocks using the multi-scale visual features from the pixel decoder. As output of each block, the queries $X^{(l)}$ generate intermediate mask and class predictions:

$$\hat{M}^{(l)} = \text{sigmoid}(X^{(l)} * F_{\text{pix}}), \quad \hat{c}^{(l)} = \text{softmax}\left[\text{Linear}(X^{(l)})\right] \tag{1}$$

where $*$ refers to pixel-wise multiplication. The final predictions are made by the queries $X^{(L)}$. Our pixel decoder, mask prediction and classification follow the design of Mask2Former [33], which handles panoptic, instance, and semantic segmentation in a unified manner. The key differences lie in 1) the transformer decoder design, and 2) the training strategy, which we detail subsequently.

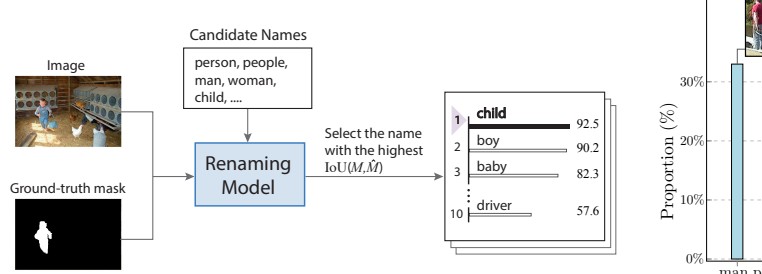
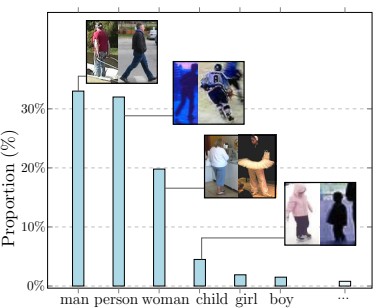

(a) Process of obtaining renovated names.

(b) Renovated name distribution within the original "person" class.

Figure 3: **Obtaining renovated names.** In (a) we illustrate how we use the renaming model to obtain a renovated name for each segment. In (b) we demonstrate that the renaming results are helpful to dataset analysis with examples from "person" class.

**Transformer Decoder.** Our transformer decoder uses text embeddings as input queries, unlike most open-vocabulary segmentation models that employ text embeddings solely as weights of the Linear layer in Eq. (1) for classification. This key change fosters an earlier integration of text and visual features at varying scales, essential for leveraging text information in segmentation tasks. As shown in Fig. 2, the text embedding of each candidate name from the segment's class "bicycle" feeds into the transformer decoder, generating their corresponding mask and class predictions.

Next, within each block of the transformer decoder, the masked cross-attention (CA) layer adopts ground-truth masks as the attention biases, and the self-attention (SA) layer [34] and feed-forward network (FFN) are the same as in Mask2Former. The queries interact with the visual features at the masked CA layers, which attend within the region defined by the ground-truth mask, i.e., the attention bias. In the example of Fig. 2, the visual features corresponding to the segment "bicycle" can therefore contribute more effectively to refining the query. To encourage the model to rely more on the textual information in prediction, we further randomly replace the ground-truth masks with predicted masks from the preceding block in the intermediate layers. Importantly, while ground-truth mask based attention biases augment the V-L interaction by focusing on the precise segment regions, they do not reduce the task to a straightforward use of ground-truth mask inputs for mask prediction, as the output queries remain a set of vectors. The final mask prediction $\hat{M}^{(L)}$ is based on the correlation of these output queries $X^{(L)}$ with the feature map $F_{\text{pix}}$ generated by the pixel decoder as in Eq. (1).

**Training Strategy.** With the CLIP backbone kept frozen, we train the transformer decoder together with the pixel decoder. To recover the ground-truth mask $M_i$ and class $c_i$, the transformer decoder makes multiple mask and class predictions with candidate names from the class $c_i$. The prediction with the highest Intersection over Union (IoU) with $M_i$ is selected for loss computation:

$$\mathcal{L}_i = \mathcal{L}_{\text{mask}}(M_i, \hat{M}_{\text{best}}) + \mathcal{L}_{\text{class}}(c_i, \hat{c}_{\text{best}}), \qquad (2)$$

where $\mathcal{L}_{\text{mask}}$ and $\mathcal{L}_{\text{class}}$ are the mask localization and classification loss functions in Mask2Former. To provide extra supervision on the name quality, we append the candidate names with a "negative" name randomly selected from a different category than the ground-truth class $c_i$. If a "negative" name scores the highest IoU, we supervise the predictions with an empty mask and the "void" class ($c_{void}$) which is one extra class in addition to the training classes:

$$\mathcal{L}_i = \mathcal{L}_{\text{mask}}(\mathbf{0}, \hat{M}_{\text{best}}) + \mathcal{L}_{\text{class}}(c_{void}, \hat{c}_{\text{best}}). \qquad (3)$$

The "void" class was also used in Mask2Former classification when the prediction mask is not matched to any ground-truth mask in the image. Having both the positive and negative supervision, we effectively incentivize the model to favor names of high quality and penalize those of low quality, thereby aiding in the accurate identification of the best-matching names for each segment.

### 3.3 Obtaining renovated names

As illustrated in Fig. 3a, to obtain the renovated names, we run the trained renaming model to associate each ground-truth mask with the candidate name. For each segment, our renaming model first

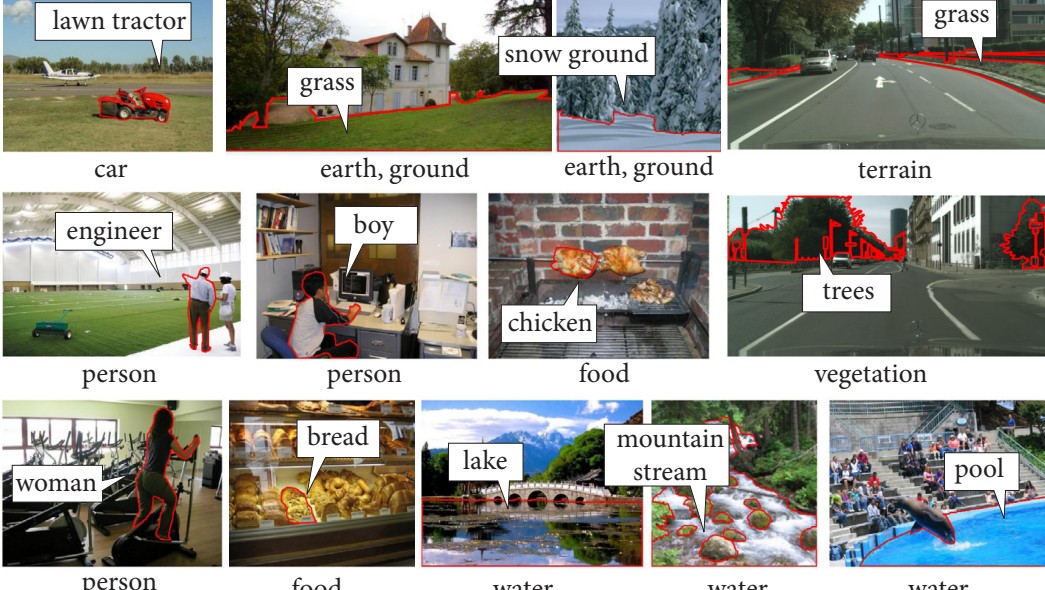

Figure 4: **Examples of renovated names on segments** from the validation sets of ADE20K and Cityscapes. For each segment, we show the original name below the image and the renovated name in the text box. See more visual results in the supplements.

ranks the names by the IoU scores between the predicted masks and the ground-truth mask, then selects the top-scoring name as the renovated name. While the IoU ranking scores are not probabilities, they are valuable to understand the relative and absolute model confidence in the names, i.e., how the chosen name stands against the other candidates and whether the selection is indeed confident. In the experiments (Section 5.3), we make use of this information to speed up human verification. We show some examples of renovated names in Fig. 4 and more in the supplements.

Our renovated names are readily usable for analyzing the dataset distribution and biases in a more fine-grained manner, as exemplified in Fig. 3b. Note that our renovated names in each original class are not necessarily mutually exclusive because different visual segments may be matched to names at different hierarchies. For example, in Fig. 4, a person is named as "engineer" (second row), likely due to the clothing, while the other person is named as "woman" (third row), possibly from the physical appearance. But "engineer" and "woman" are not mutually exclusive names. An interesting future extension would be to further study the hierarchies of different names in a dataset.

## 4 Applications of RENOVATE Names

We further demonstrate the value of the renovated names with two application tasks: training stronger open-vocabulary segmentation models and improving existing evaluation benchmarks. The improvements in both tasks indicate the quality of RENOVATE names.

### 4.1 Training with RENOVATE names

RENOVATE names offer precise and diverse text information. We exploit this to improve model training and achieve better generalization and data efficiency. Following prior works [24, 25, 35], we replace original ground-truth names with RENOVATE names to compute text embeddings. In addition, we employ the "negative sampling" technique, commonly used in natural language processing to enhance training with a large number of classes [36, 37, 38, 39]. For each segment, we first randomly select $C - 1$ negative classes and choose one RENOVATE name from each. These names are then concatenated with the ground-truth RENOVATE name to form the text prompt. This effectively constructs text embeddings of length $C$ with semantically distinct names, improving training efficiency and generalization. For computing the classification loss, we use the cross entropy function on a per-segment basis, i.e., for an image with segments $\{m_i\}_{i=1}^N$, $\mathcal{L}_{\text{class}} = \sum_{i=1}^N \text{CE}(c_i, \text{softmax}(v_i T_i^T))$, where $c_i$ is the ground-truth label, $v_i \in \mathbb{R}^d$ is the predicted

visual embeddings and $T_i \in \mathbb{R}^{C \times d}$ is the constructed text embeddings through negative sampling. Throughout our experiments, we use negative sampling in our model training unless specified.

## 4.2 Improving evaluation with RENOVATE names

RENOVATE names also enable more fine-grained analysis of open-vocabulary models. For instance, for a misclassification from "car" to "van", we may distinguish between "SUV" to "minivan" versus "sedan" to "delivery van", which isn't possible with class-level annotations in current datasets. However, even with RENOVATE names, standard segmentation metrics obscure our desired fine-grained performance insights since they penalize all such misclassification cases equally. To this end, we adopt "open" evaluation metrics [40] that consider semantic similarity between names. Specifically, for a groundtruth mask pixel $g_i$ with label $c_i$ and a predicted mask pixel $d_j$ with predicted label $c_j$, we compute soft $TP_{c_i} = S_{c_i,c_j}$, soft $FP_{c_j} = 1 - S_{c_i,c_j}$, and soft $FN_{c_i} = 1 - S_{c_i,c_j}$, where $S_{c_i,c_j}$ is the semantic similarity between $c_i$ and $c_j$. These soft metrics are then used to compute panoptic quality (PQ), Average Precision (AP), and mean IoU (mIoU) in a standard way. In Section 5.3, we demonstrate how open metrics effectively reveal that there are indeed many "benign" misclassification scenarios using RENOVATE names.

While our renaming process is automated, it is essential to ensure exceptionally high-quality names for evaluation. Therefore, we introduce a rigorous verification process involving five human annotators who review the top three name suggestions and their confidence scores from our model to select the best match. If none of the top three names are suitable, verifiers examine the entire candidate list. In case of disagreements, verifiers discuss and decide collectively. Note that human verification is used only for evaluation sets, while automatically generated names are used for the much larger training sets. Future work could explore using multiple vision-language foundation models to automate this verification process as evaluation sets grow [41, 42, 43].

# 5 Experiments

## 5.1 Obtaining renovated names

**Setup.** We renovate three panoptic segmentation datasets respectively: MS COCO [13], ADE20K [14], and Cityscapes [15]. To generate context names, our default image captioning model is CaSED [32], which retrieves captions by matching vision embeddings with text embeddings of captions from a large-scale PMD dataset [44]. We train a renaming model for 60k iterations with a batch size of 16 on the training set, then generate RENOVATE names for the entire dataset.

**Results.** Table 2 summarizes the statistics of RENOVATE names. Compared to the original class names, each dataset has approximately 5-6 times more distinct RENOVATE names, indicating that they provide more diverse and fine-grained semantic annotations to the visual segments. We show some visual examples in Fig. 4 and more in Appendix E.1. To further validate

|  | COCO | ADE | CS |
|---|---|---|---|
| # Original classes | 133 | 150 | 19 |
| # Segments/Class | 11,274 | 2,009 | 4,964 |
| # RENOVATE names | 741 | 578 | 108 |
| # Segments/Name | 1,871 | 521 | 873 |

Table 2: **Statistics of renovated datasets.**

the quality of the names, we conduct a human preference study, see Appendix B.1. We also conduct an ablation analysis on the components of the renaming pipeline, see Appendix B.2.

## 5.2 Training with renovated names

**Setup.** We train FC-CLIP [25] models on MS COCO with our renovated segment-matched names and compare them with models trained using other name sets on COCO, ADE20K, and Cityscapes. Specifically, "OpenSeg" names [5] were manually curated by previous researchers. "Synonym names" are generated by prompting GPT-4 based on the original class names. "Candidate names" are the outputs of the initial step in our renaming process, using image captioning to provide context for prompting GPT-4. A key difference between these names and RENOVATE names is that the former uses a set of names for a class of segments, whereas RENOVATE names provide a name for each individual segment. For model evaluation, we follow prior practices [5, 24, 25] to group

| Train names | Source dataset | | | Target datasets | | | | | |
| | MS COCO | | | ADE20K | | | Cityscapes | | |
| | PQ | AP | mIoU | PQ | AP | mIoU | PQ | AP | mIoU |
|---|---|---|---|---|---|---|---|---|---|
| Original names | 52.70 | 42.72 | 60.91 | 25.61 | 15.97 | 33.06 | 46.06 | 23.67 | 57.99 |
| OpenSeg names | 53.60 | 43.87 | 60.56 | 27.14 | 17.41 | 35.63 | 45.34 | 24.05 | 58.42 |
| Synonym names | 23.38 | 35.87 | 47.20 | 15.56 | 14.78 | 29.74 | 22.94 | 19.58 | 45.91 |
| Candidate names | 41.32 | 40.96 | 56.68 | 19.80 | 15.85 | 33.43 | 39.20 | 20.79 | 51.80 |
| RENOVATE | **56.62** | **45.50** | **65.48** | **27.98** | **17.94** | **38.50** | **46.10** | **27.99** | **61.25** |

Table 3: **Training with renovated names.** During inference, test names are merged from Original, OpenSeg, and RENOVATE names for fair comparison. Our results demonstrate that RENOVATE names can help train stronger open-vocabulary models.

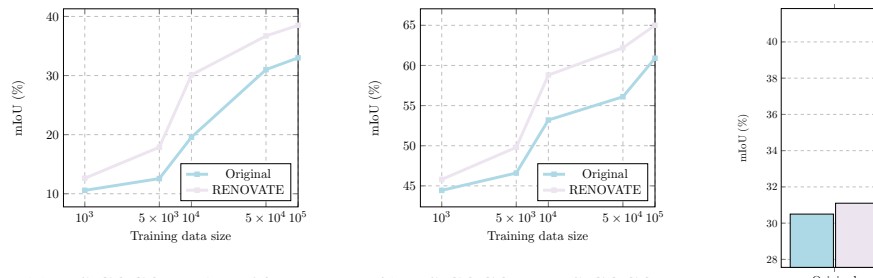

(a) MS COCO → ADE20K.   (b) MS COCO → MS COCO.

Figure 5: **Data efficiency comparison between RENOVATE and original names.**

Figure 6: **Effectiveness of negative sampling.**

predictions based on the original class divisions. For a fair comparison, our test name set merges names from Original, OpenSeg, and RENOVATE names.

**Results.** Our results in Table 3 demonstrate that using RENOVATE names for training improves segmentation on both source dataset and target datasets. For example, compared to the Original names, the model trained with RENOVATE names improves by near 4 PQ on MS COCO and over 5% mIoU on ADE20K, showing significantly larger gains offered by OpenSeg names. This underscores the value of name refinement on per-segment level and indicates the high quality of RENOVATE names. We also note that models trained with Synonym names and Candidate names show inferior performance even to the Original names, reflecting the importance of high name quality.

The richer text information offered by RENOVATE names also improves training efficiency. In Fig. 5, we show that models trained with RENOVATE names can achieve comparable performance with significantly less data compared to those trained with the Original names. For instance, models trained with $10^4$ images using RENOVATE names can match or exceed the performance of models trained with $5 \times 10^4$ images using the Original names. This corroborates previous findings that improving data quality can reduce the required quantity of training data [45, 46].

**Ablation on the negative sampling.** In Fig. 6, we study the effectiveness of negative sampling. Interestingly, we found that negative sampling is especially helpful to RENOVATE names while minorly impacting models trained with original names. This is because RENOVATE names have much more class names and the names are also more semantically close. Negative sampling effectively controls the number of classes used in cross entropy calculation and keeps the name embeddings relatively distinct, thus significantly improving training effectiveness for RENOVATE names.

## 5.3 Improving evaluation with renovated names

**Setup.** We evaluate pre-trained FC-CLIP [25], MasQCLIP [35], and ODISE [24] models on ADE20K and Cityscapes using three different name sets with both standard and open metrics. For open metric computation, we need the similarity between the ground-truth and predicted class names. For Original and OpenSeg names, which use a set of names for a class of segments, we use the averaged pairwise name similarity between classes. For RENOVATE names, we can directly use

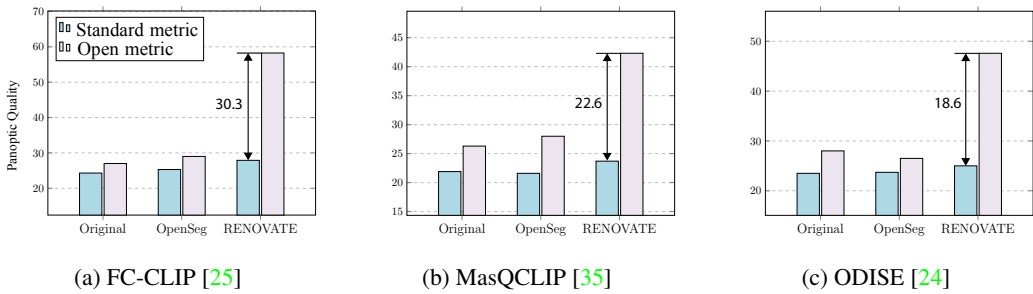

(a) FC-CLIP [25]   (b) MasQCLIP [35]   (c) ODISE [24]

Figure 7: **Open-vocabulary evaluation on ADE20K with different names, metrics, and models.**

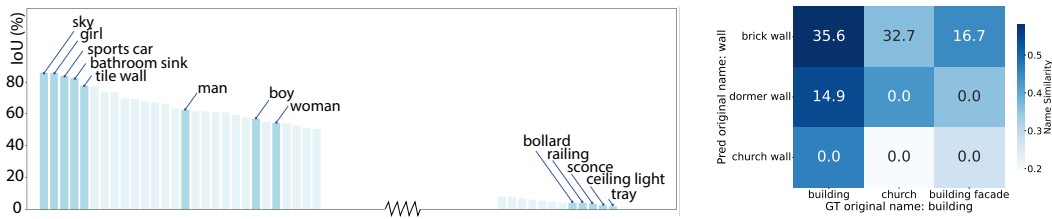

(a) Per-category IoU on the ADE20K with 578 RENOVATE classes.   (b) Misclassification analysis.

Figure 8: **RENOVATE names enable more fine-grained analysis on models.** (a) Per-category IoU with highlighted top/bottom-5 RENOVATE names and selected names from the "person" class in ADE20K. (b) Confusion matrix of frequently misclassified RENOVATE names from "building" to "wall", showing misclassification proportions (numbers) and pairwise name similarity (color).

the pairwise name similarity between renovated names. Fig. 7 shows PQ evaluation of pre-trained models on ADE20K. Full evaluation results, including mAP and mIoU and results on Cityscapes are provided in Appendix B.3.

**Results.** In Fig. 7, we first note that the discrepancy between standard and open metrics is significantly greater when evaluated using RENOVATE names. This indicates that many misclassification cases are indeed "benign", *i.e.*, when evaluated using more fine-grained names, the misclassified names turned out to be semantically close to the ground-truth names. For example, when an "area rug" is predicted to be a "floor rug" (RENOVATE names), it will be penalized much less than from "rug" to "floor" (original names); but if an "area rug" is predicted as "concrete floor", the classification error will be even higher. This shows that current pre-trained models have learned relatively good semantic concepts, while still being sensitive to the text prompts at inference time.

RENOVATE names also enable a more fine-grained analysis of the models. In Fig. 8a, we present a per-category IoU analysis on RENOVATE names within the ADE20K dataset. We can see that large, well-defined objects such as "sky" and "girl" have high IoU scores, while small, deformable, and rare classes like "bollard" and "ceiling light" are more challenging for current models. Note that most of these names do not exist in the original benchmarks. In addition, our analysis reveals model biases, evidenced by the disparity in IoU scores between "man" and "woman" as well as "girl" and "boy", suggesting an imbalance in the training dataset. In Fig. 8b, we further illustrate how RENOVATE names facilitate a more detailed investigation of misclassifications. For example, among "building" segments misclassified as "wall", 32.7% "churches" from the "building" class are predicted to be "brick wall" from the "wall" class. Additionally, we observe a strong correlation between name similarity and misclassifications. This showcases that our RENOVATE names can help identify problematic sub-categories within the original class that models still struggle with.

Finally, we note that the open metrics are penalizing more on misclassified concepts with lower semantic similarity, e.g., "car/road" mistakes are penalized more than "car/truck" mistakes. In other use scenarios, different types of mistakes (e.g., outlier patterns) may be preferred to be penalized more and the evaluation metrics should be desgined accordingly. However, regardless of the design of the evaluation metric, RENOVATE names make it possible to conduct fine-grained analysis on mistakes with different semantic distances. Without RENOVATE names, we can only see the coarse misclassifications without a fine-grained understanding of the model.

# 6 Conclusion and Limitations

In this work, we address the naming issues and show that renaming improves both model training and evaluation of open-vocabulary segmentation. While RENOVATE uncovers model biases, it could inadvertently propagate biases from the foundational models into the new names. To mitigate potential negative societal impacts, we advocate for verification of names in critical applications, as exemplified by our verification of names in evaluation sets. As the first attempt to propose a generalized renaming framework, we acknowledge that our exploration remains incomplete. In the future, we aim to further refine our method, exploring more design choices such as other model backbones [47], and scale it up to the publicly available large-scale datasets [48, 49].

## Acknowledgement

Haiwen Huang and Dan Zhang were supported by Bosch Industry on Campus Lab at University of Tubingen. Andreas Geiger was supported by the ERC Starting Grant LEGO-3D (850533) and the DFG EXC number 2064/1 - project number 390727645. Haiwen Huang would like to thank Tianlin Ye for her help in figure making.

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

## Renovating Names in Open-Vocabulary Segmentation Benchmarks Supplementary Material

This supplementary material to the main paper "Renovating Names in Open-Vocabulary Segmentation Benchmarks" is structured as follows:

- In Appendix A, **More Literature Review**, we provide a brief overview of works that address name problems in open-vocabulary classification.

- In Appendix B, **More Experiment Results**, we provide more experiment results, including a human preference study on the name quality, additional ablation analysis on the renaming pipeline, full results on improving evaluation using renovated names and some more examples of using RENOVATE names to conduct fine-grained misclassification analysis.

- In Appendix C, **More Implementation Details**, we elaborate on the implementation details on the GPT-4 prompts for name generation, human verification process for RENOVATE names on evaluation sets, open metrics, and the renaming and pre-trained models.

- In Appendix D, **More Name Examples**, we present more examples of context, candidate, and renovated names. We also show more examples of name selection with the IoU scores.

- In Appendix E, **More Qualitative Analysis**, we provide more qualitative analysis demonstrating that RENOVATE can refine original names, correct wrong annotations, and uncover segments with shared semantic concepts across datasets.

- In Appendix F, **License for Existing Assets**, we list the names of the license for each assets used in this paper.

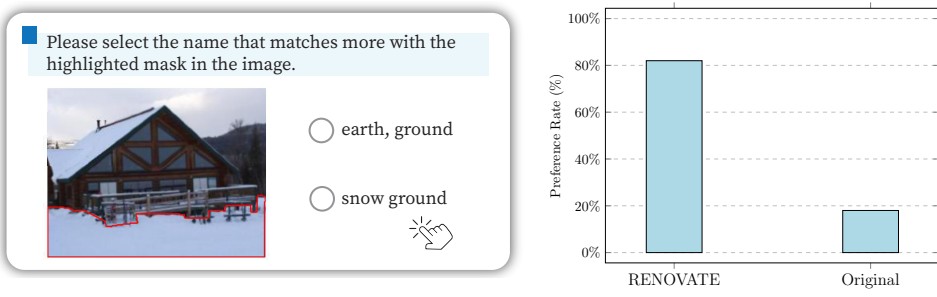

(a) User interface for human preference study.       (b) Human preference results.

Figure B.1: **Human preference study.** A survey of 20 researchers is conducted to compare preferences between the original names versus RENOVATE names on the validation sets. RENOVATE names are favored in 82% cases.

## A More Literature Review

In this section, we provide a brief overview of name problems in open-vocabulary classification. While OpenSeg [5] is the only prior work for the name problems in open-vocabulary segmentation, there are more research works on improving name quality for open-vocabulary *classification*. For example, [50] improve class names by including their parents and children from the WordNet hierarchy [51]. [32] propose to replace the fixed vocabulary list with nouns extracted from best-matching captions. [52] improve the descriptiveness of the vocabulary by adding class descriptions. However, since the VLMs used by these models are all trained on image-text datasets, it is non-trivial to extend them to dense prediction tasks like segmentation. For the renaming strategy, our context name design and LLM-based generation approach ensure the diversity of our names with hierarchical concepts as well as descriptive contexts, without the use of an external word hierarchy as [51] or class descriptions as [52]. Our renaming model further improves the name quality based on the visual segments and is more specifically designed for the segmentation tasks.

## B More Experiment Results

### B.1 Human preference study

As demonstrated in Fig. B.1, we study the name quality of our renovated names with a human preference study on ADE20K and Cityscapes datasets, which consists of 30249 and 14670 segments in the validation set, respectively. Specifically, we first randomly select 10% of all the segments in the validation set of each dataset. The average time for human experts to finish the preference study is about 2 hours. Then for each segment, we ask 20 researchers unfamiliar with the project about their preference between its original name and our model-selected name. Our results demonstrate that our renovated names are preferred in 82% cases, showing a clear advantage over the original names. We provide more details of our human study in the supplements.

### B.2 Ablation analysis on the renaming pipeline

To conduct ablation study on the renaming pipeline, we leverage pre-trained open-vocabulary model models as evaluators of names, automating the name quality evaluation process. Instead of involving humans, we prompt pre-trained open-vocabulary segmentation models with different names. Names are then considered to be better if they help the model achieve better open-vocabulary segmentation. Specifically, we use pre-trained FC-CLIP model for the following experiments.

In Table B.1, we first analyze the impact of context names for candidate generation. Our results indicate significant performance improvements when context information is provided, as opposed to the "None" approach, which relies only on original names. This highlights the effectiveness of leveraging contextual insights for generating superior candidate names. In comparison, CaSED outperforms both BLIP2 and RAM, achieving a greater boost in performance through its retrieval-

| Context name source | ADE20K | | |
| --- | --- | --- | --- |
| | PQ | AP | mIoU |
| None | 25.50 | 16.53 | 33.80 |
| Caption nouns (BLIP2 [53]) | 25.89 | 17.36 | 34.74 |
| Image tags (RAM [54]) | 26.95 | 17.55 | 35.54 |
| Caption nouns (CaSED [32]) | **27.90** | **17.88** | **37.05** |

Table B.1: **Ablation on the context name sources.**

| Attn bias | Query | Classifier | ADE20K | | |
| --- | --- | --- | --- | --- | --- |
| | | | PQ | AP | mIoU |
| gt mask | learnable | text | 21.09 | 14.91 | 26.96 |
| gt mask | text | learnable | 25.65 | 16.33 | 32.67 |
| rand gt mask | text | learnable | 26.21 | 17.50 | 35.62 |
| rand gt mask | text w/ neg | learnable | **27.90** | **17.88** | **37.05** |

Table B.2: **Ablation on renaming model architecture.**

| GPT4 prompt | ADE20K | | |
| --- | --- | --- | --- |
| | PQ | AP | mIoU |
| full prompt | **27.9** | **17.9** | **37.1** |
| - context names | 25.5 | 16.5 | 33.8 |
| - suggestions on name types | 25.6 | 16.8 | 34.0 |
| - instructions on original names | 26.5 | 17.3 | 35.2 |

Table B.3: **Ablation on GPT-4 prompts.**

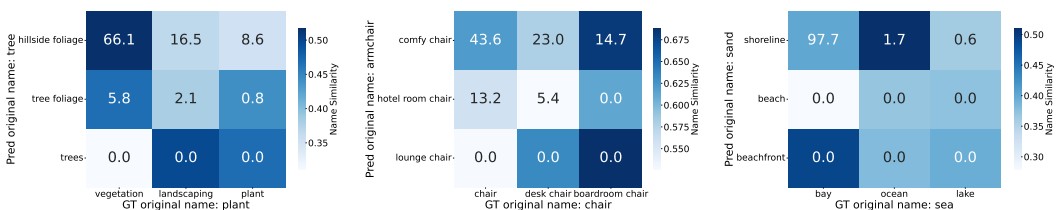

Figure B.2: **More examples of fine-grained misclassification analysis.**

based approach and the utilization of expansive external image-caption datasets, underlining the benefits of contextual enrichment in name generation.

In Table B.2, we analyze the design choices of our renaming model. We start by establishing that our approach using text queries is superior to a standard text classifier, which only interacts with the transformer decoder's output query embeddings and treats the mask prediction branch in a class-agnostic way. In comparison, our text queries interact intimately with spatial details from image features and exploit the mask prediction branch, leading to more precise name-segment matching. We also observe performance benefits from randomly substituting ground-truth masks with predicted ones in attention biases ("rand gt mask"), which suggests training with increased reliance on text queries refines the matching quality. Furthermore, augmenting our model with negative name appending ("text w/ neg") enhances its discriminative capacity, further boosting performance and solidifying the renaming model's overall effectiveness.

In Table B.3, we ablate the contribution of the different components in our GPT-4 prompts (see Appendix C.1). Specifically, our GPT prompts have the following components: (1) original name; (2) context names from image captioning; (3) suggestions on the two types of names (synonyms and short contexts) to focus, with examples; (4) instructions on how to make use of the original name, with examples. From our empirical studies, we find that GPT-4 is typically robust to the specific wording of the four components and that removing any key component (esp. (2) and (3)) can significantly affect the results.

## B.3 Improving evaluation benchmarks with renovated names

In Table B.4, we show the full evaluation results of the experiments described in Section 5.3. With results on ADE20K and Cityscapes with PQ, AP, and mIoU evaluation, we make similar observations as Section 5.3 that the performance gap between standard and open metrics is significantly larger when evaluating using RENOVATE names, indicating that many misclassification cases are indeed "benign". We also provide more examples of fine-grained misclassification analysis in Fig. B.2.

| Model | Benchmark Names | Metric | ADE20K [14] | | | Cityscapes [15] | | |
|---|---|---|---|---|---|---|---|---|
| | | | PQ | AP | mIoU | PQ | AP | mIoU |
| ODISE | Original | Standard | 21.88 | 13.94 | 29.16 | 39.72 | 27.73 | 49.60 |
| | | Open | 26.31 | 14.49 | 34.39 | 44.09 | 27.91 | 58.20 |
| | OpenSeg | Standard | 21.63 | 13.95 | 28.88 | 43.26 | 28.45 | 54.53 |
| | | Open | 28.02 | 15.32 | 38.86 | 46.66 | **28.88** | 58.61 |
| | RENOVATE | Standard | **23.69** | **14.38** | **31.64** | **43.61** | 28.38 | **57.42** |
| | | Open | **42.32** | **50.11** | **44.65** | **53.10** | **34.52** | **65.05** |
| MasQCLIP | Original | Standard | 23.46 | 12.80 | 30.32 | 33.78 | 18.11 | 45.35 |
| | | Open | 26.53 | 13.01 | 36.33 | 37.94 | 18.35 | 55.75 |
| | OpenSeg | Standard | 23.70 | 12.84 | 31.17 | 35.05 | 17.79 | 46.46 |
| | | Open | 28.04 | 13.89 | 39.40 | 38.83 | 18.62 | 52.11 |
| | RENOVATE | Standard | **25.00** | **12.93** | **32.30** | **35.63** | **18.19** | **50.07** |
| | | Open | **47.59** | **67.01** | **42.75** | **49.30** | **37.25** | **61.44** |
| FC-CLIP | Original | Standard | 24.30 | 16.79 | 32.14 | 39.42 | 22.76 | 53.08 |
| | | Open | 27.04 | 16.52 | 36.59 | 42.57 | 26.54 | 62.69 |
| | OpenSeg | Standard | 25.35 | 17.30 | 33.06 | 43.68 | 26.93 | 56.15 |
| | | Open | 29.03 | 17.52 | 41.03 | 45.92 | 27.09 | 60.21 |
| | RENOVATE | Standard | **27.90** | **17.88** | **37.05** | **45.90** | **29.79** | **62.55** |
| | | Open | **58.25** | **66.99** | **48.31** | **58.63** | **38.91** | **70.60** |

Table B.4: **Open-vocabulary segmentation on using both standard and open metrics across different pre-trained models using different name sets.**

# C  More Implementation Details

## C.1  GPT-4 prompts for name generation

We use GPT-4 ChatCompletion API [55] for candidate name generation. For each class, we provide two messages. The first message, i.e., "system message", describes the role of the system, where we describe our overall problem setup and goals. The second message, i.e., "user message", describes the intention of the user, where we provide the original class names and context names. We show our exact prompt messages as follows:

**System message:** You are a helpful assistant aiding in renaming dataset classes. Each class has an inadequate original name and a set of *context names* derived from related captions (with their frequencies sorted and listed in brackets). These context names provide insights into the category's essence. When renaming, you may: *1. Use synonyms or subcategories of the original class name* (e.g., 'grass' can be renamed as 'lawn, turf'). 2. *Provide a short context to address polysemy* (e.g., 'fan' can be renamed as 'ceiling fan, floor fan'). Please generate new names for each class in lower case, listed in a row. Ensure the new names logically connect to the original class, using it as the head noun. Avoid arbitrary noun concatenations and nonsensical names. For instance, the class 'sky' should not yield names like 'person under sky'. Ready to proceed with naming? Kindly provide the original class names and context names.

**User message:** Original name: {original class name}, context names (with frequencies) are {context names}. What are the new names? Only provide 5-10 names. And make sure you generate at least 3 reasonable synonyms or subcategories.

For the system message, we start the message with "You are a helpful assistant" following the examples in OpenAI Chat Completions API [55]. We then describe our problem setup, input format, and instructions with examples on name generation. For the user message, {original class name} and {context names} require inputs from each category. For our ablation experiments without context names ("None" in Table B.1 in the main paper), we remove the descriptions about context names in the system message and do not require context names in the user message.

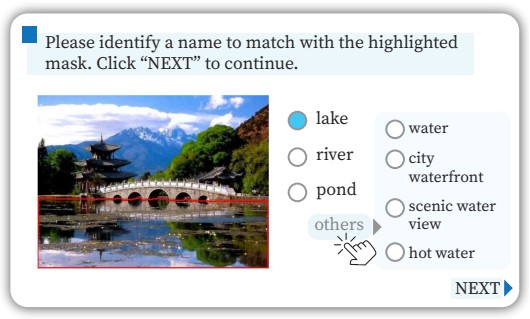
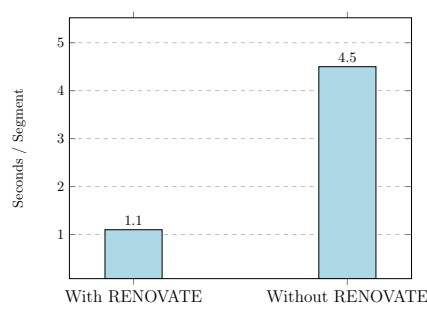

|  (a) User interface for name verification. | (b) Comparison of verification speed. |

Figure C.1: **Human verification for upgrading benchmarks.** We ask 5 human annotators to verify the names of segments in the validation sets of ADE20K and Cityscapes. As shown in (a), humans are given the top 3 suggestions from the model and are asked to verify the selected name or choose a more matching name from either the other top 3 names or the rest of the candidate names. (b) shows that RENOVATE name suggestions significantly speed up the human verification/annotation process.

## C.2 More details on the human verification process

To ensure the high quality of names on the evaluation sets of ADE20K and Cityscapes, we verify the names on all the segments. As shown in Fig. C.1a, for each segment, we provide the model-selected name along with the other top 3 names ranked by the model, we also provide the rest of the candidate names in the "others" option. Therefore, the human verifier can quickly continue to the next segment if the selected name itself or another name in top 3 names matches the most with the segment. If none of the top 3 names fits well, humans need to choose from the other candidate names in "others". In the latter case, it would be similar to the scenario where RENOVATE suggestions are not provided and humans have to choose from the whole un-ranked candidate name list. In Fig. C.1b, we compare the name verification speed with and without RENOVATE suggestions on 200 segments. As we can see, RENOVATE suggestions significantly speed up our verification process. We note that the human verifiers are provided only with candidate names from the same original class, which is typically the original ground-truth class. In the rare cases where our renaming model prefers names from classes other than the original groundtruth class, we first verify the class choices, then provide the human verifiers with the name suggestions from the verified class.

We further note that for all the segments verified, 68% of them are matched to the default, i.e., top 1, model-selected name, and 92% of them are matched to the top 3 RENOVATE suggestions, reflecting the high quality of RENOVATE names. In Fig. C.2, we show typical cases when humans and models disagree, including segments that are too small to recognize and segments whose names cannot be inferred correctly due to insufficient visual cues. We note that such segments only consist a minority of all the segments, with inherent difficulty to name precisely.

## C.3 More details on the implementation of open metrics

To compute the open metrics [40], we need a similarity function on names. Previous research [40] compared three types of similarity measures: WordNet methods [56, 57, 58], text embedding models [36, 59, 60], and pre-trained language models [23, 61, 62, 63]. They demonstrated that both WordNet methods and text embedding models compute reasonable similarity measures while pre-trained language models tend to overestimate similarity between names. However, since the names in our work are in free form and may not map to any synonym sets in WordNet [51], we adopt one of the text embedding models, FastText [60], which is shown to compute reasonable similarity measures [40].

## C.4 Other implementation details

**More details on the renaming model.** For the backbone of the renaming model, we use ConvNeXt-Large CLIP [23, 64] from OpenCLIP [64] pretrained on LAION-2B dataset [65]. Following

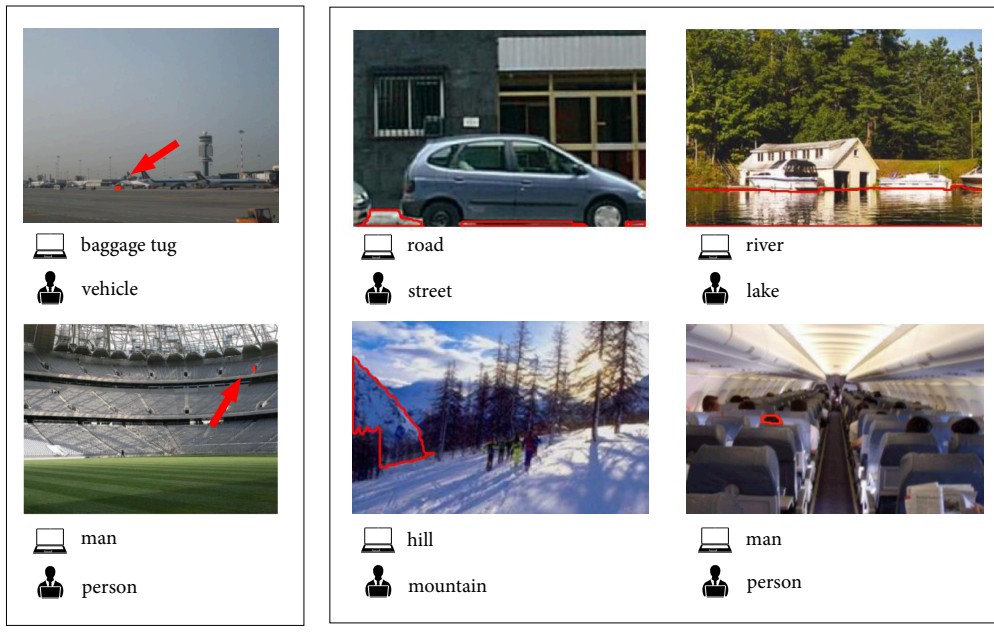

(a) Segments are too small.  (b) Segments lack sufficient visual cues.

Figure C.2: **Typical cases when humans and models disagree during human verification.** We show two typical cases when humans may choose different names from our model-selected names. (a) shows when segments are too small to recognize, it can be difficult to decide which name is correct. This may indicate a limitation of our renaming method on extremely small objects. (b) shows when there is a lack of sufficient visual cues to infer the names of the segments, the choices are ambiguous. For example, both "river" and "lake" are reasonable choices for the segment in the top right image without more information of the scene. Note (a) and (b) consist only a minority of all the segments.

Mask2Former [33], our transformer decoder consists of nine blocks and we use AdamW [66] optimizer with a learning rate of $10^{-4}$ and a multi-step decay schedule with weight decay of 0.05. Training one renaming model on 4 A-100 GPUs requires approximately 3 days.

**More details on training with renovated names..** We follow the training schedule in prior works [25] to train FCCLIP and use negative sampling with $C = 100$ in training all models. Training one model on 4 A-100 GPUs requires approximately 4 days.

**More details on the pretrained models.** ODISE [24] provides two pretrained models, ODISE (label) and ODISE (caption), trained on category names and caption nouns respectively. We use ODISE (label) since they demonstrated stronger performance than the other model in the original paper. For ODISE and FC-CLIP, we set their geometric mean parameters $\alpha = \beta = 0.5$ so we do not differentiate the treatment of test names based on their name overlapping with the training names. In this way, we can compare different test names in a fair way. By default, we use FC-CLIP for all of our ablation studies. For the test prompt template, we follow prior work [9, 35, 25] to use the ViLD template [9], which consists of 63 prompt templates such as "There is {article} {category} in the scene." These templates are ensembled by averaging the text embeddings.

## D    More Name Examples

In Table D.1, we show examples of original names, context names, candidate names, and RENO-VATE names. For RENOVATE, the names for each class are simply the set of unique names for all the instances in that class after renaming. We can see that the context names are important for understanding the vague class names such as "earth" and "field" and are helpful for providing more

| Original name | Context names | Candidate names | Renovated names |
|---|---|---|---|
| earth, ground | building, tree, stand, water, person, floor, sky, road, snow, car | ground cover,earth floor,ground cover, ground,earth surface,snow ground, land,natural ground,outdoor ground,terrain, earth,dirt ground,terrestrial terrain | ground cover, terrestrial terrain, natural ground, snow ground, dirt ground, ground, grass |
| field | lush, field, sky, green, grass, tree, road, hillside, grassy, rural | rural field, roadside field, green field, crop field, sports field, grassland, grassy hillside, | rural field, roadside field, grassland, crop field, sports field |
| rock, stone | stone, water, tree, lush, sea, sky, shoreline, coast, creek, stand | rock,stone,standing stone,beach pebble, shoreline rock,national park boulders, river stone,stones,lush stone,rocks | river stone, stones, beach pebble rock, national park boulders, shoreline rock |
| fountain | city, fountain, building, garden, square, park, water, hotel, place, design | garden fountains,fountain,urban fountains, square fountains,park fountains, water feature designs,hotel water features | hotel water features, urban fountains,park fountains |
| counter | room, area, lobby, reception, shop, store, hotel, design, office, interior | checkout counter,reception desk, office counter area,countertop,hotel lobby counter, cashier counter,counter,retail store counter, service desk,storefront counter,information kiosk | information kiosk, reception desk, cashier counter, counter, office counter area, checkout counter |
| stairway, staircase | building, stair, floor, room, ceiling, lead to, pillar, balustrade, stairwell, rail | balustrade staircase,ceiling stairwell,stairway, building stairs,staircase,pillar-supported stairs, room stairway,railing stairway,floor staircase | pillar-supported stairs,railing stairway, room stairway, floor staircase |
| bar | bar, restaurant, room, area, hotel, interior, pub, table, house, counter | bar table,bar area,restaurant bar,hotel bar, home bar,bar counter,bar interior,bar,pub bar | bar area, bar table bar counter |
| canopy | room, bed, bedroom, house, idea, design, palace, garden, home, royal | home canopy,canopy,house canopy, garden canopy,royal palace canopy,patio canopy, bedroom canopy,room canopy | bedroom canopy, patio canopy, garden canopy |
| grandstand, covered stand | stadium, game, field, arena, world, park, football, school, sport, high | high-capacity grandstand, park bleachers, covered game stand, world-class viewing area, spectator stand, football field seating,grandstand, sports arena stand,stadium grandstand, school spectator seats | sports venue grandstand, sports arena stands, park bleachers, spectator stand football field seating |
| swimming pool, swimming bath, natatorium | pool, swimming, house, hotel, villa, resort, beach, indoor, luxury, apartment | luxury resort pool,swimming bath,villa pool, apartment complex pool,hotel swimming pool, home swimming pool,swimming pool, beachfront swimming pool,indoor swimming bath | hotel swimming pool, beachfront swimming pool, villa pool, indoor swimming pool |

Table D.1: **Examples of context names, candidate names, and RENOVATE names for classes from ADE20K.**

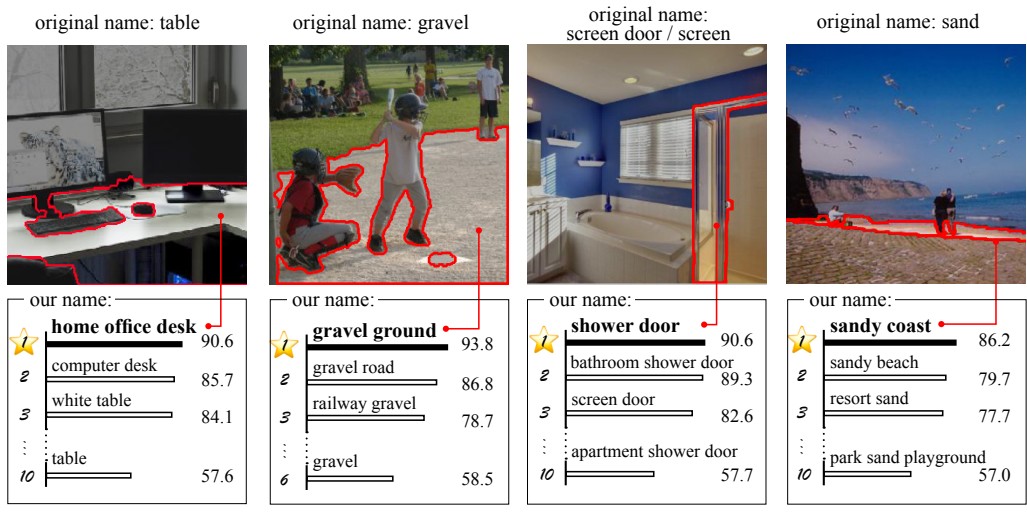

Figure D.1: **More examples on name selection based on IoU scores.**

contexts to names with multiple potential meanings like "counter" and "bar". We can also see that RENOVATE names provide more comprehensive descriptions of the corresponding classes.

In Fig. D.1, we further show more visual examples of the name selection process (Section 3.3). Together with Fig. 3 in the main paper, these examples demonstrate that our renaming model provides a meaningful ranking of the names to match with each segment.

# E  More Qualitative Analysis

## E.1  RENOVATE can refine original names

In Fig. E.1 and Fig. E.2, we present more examples of RENOVATE names along with the original names of segments from ADE20K and MS COCO. We show that RENOVATE refines the original names by being more precise and closer to human natural language.

## E.2  Other potential use cases of RENOVATE

**RENOVATE can correct annotation errors.** From Fig. E.3, we can see that our renaming model discovers wrong annotations and finds more accurate names through RENOVATE. It's important to highlight that these renovated names do not belong to the original ground-truth class; instead, our model ranks them over names from the original class. This demonstrates the renaming model's utility in detecting and correcting inaccuracies in annotations by effectively matching names with visual content.

**RENOVATE uncovers segments with shared semantic concepts across datasets.** Since RENOVATE aims to describe visual segments using more human-aligned language, it effectively standardizes the naming conventions for identical concepts in different datasets. In Fig. E.4, we demonstrate this with examples from MS COCO and ADE20K. While renaming each dataset separately, we observe that RENOVATE inadvertently reveals the shared name spaces by identifying more appropriate names in natural language to describe the visual contents. This property can be potentially used for analyzing the differences between datasets, merging datasets to construct new benchmarks, and various other data curation processes.

**RENOVATE can rename segments in an image when combined with other foundation models.** In Fig. E.5, we showcase another possible application of our renaming model by further applying it on generated masks from SAM2 model [67] (points per side=12) and generated image tags from the Recognize-Anything-Model (RAM) [49]. Specifically, we use the generated masks as attention biases and each mask is paired with the full list of the generated tags as the text queries. The renaming model is used to obtain the best-matching name for each mask. The good mask-name matching performance demonstrates that our renaming model can potentially be used on datasets where mask annotations are not available.

# F  License for Existing Assets

- CaSED code and model [32]: MIT License.
- GPT-4 model [31]: https://openai.com/policies/terms-of-use/
- FC-CLIP code and model [25]: Apache License 2.0.
- ODISE code and model [24]: Custom license (https://github.com/NVlabs/ODISE/blob/main/LICENSE)
- MasQCLIP code and model [35]: Custom license (https://github.com/mlpc-ucsd/MasQCLIP/blob/main/LICENSE)
- MS COCO [13]: Creative Commons Attribution 4.0 License.
- ADE20K [14]: Creative Commons BSD-3 License Agreement.
- Cityscapes [15]: Custom licence (https://www.cityscapes-dataset.com/license/)

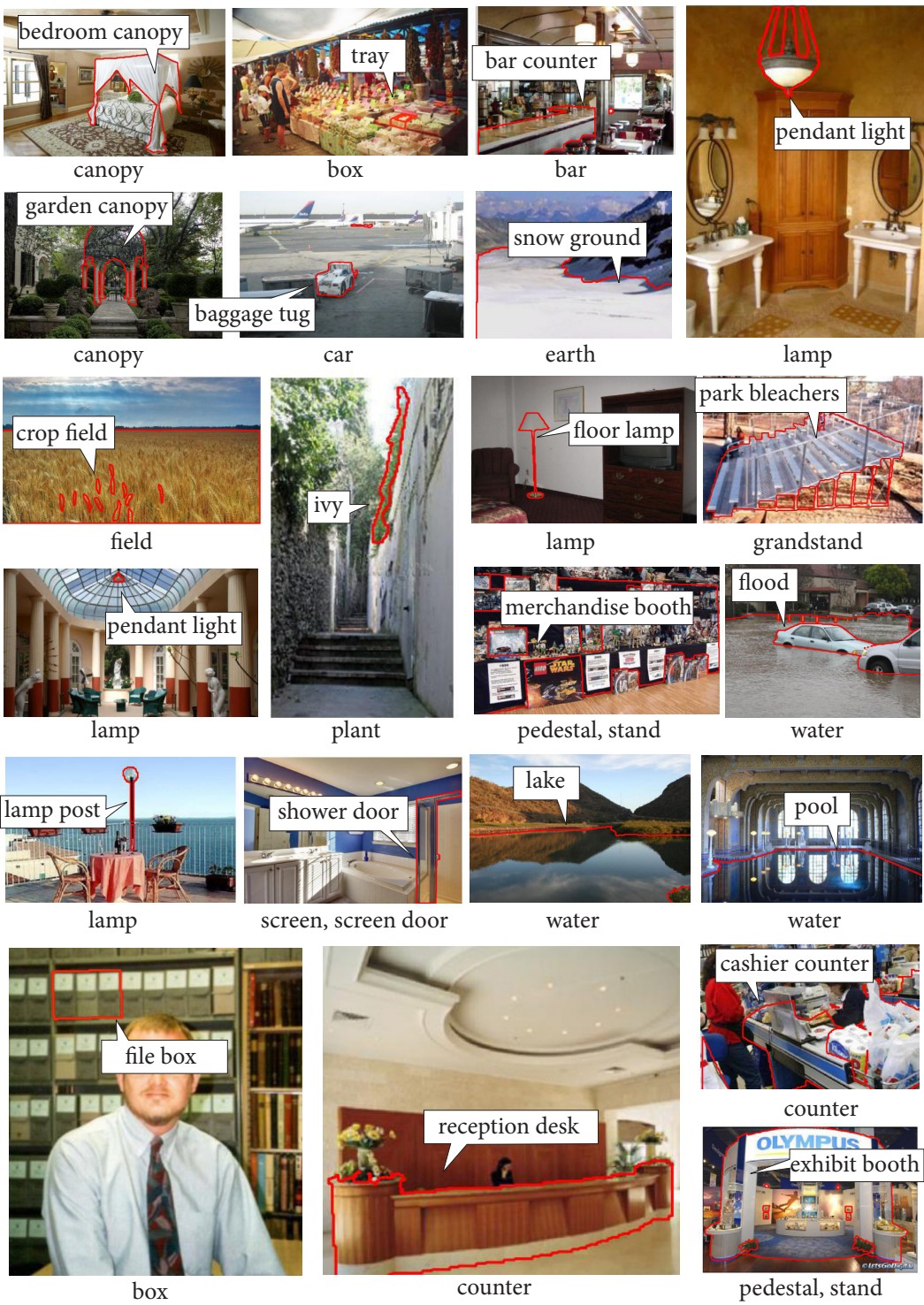

Figure E.1: **More examples of renovated names on segments** from the validation set of ADE20K. For each segment, we show the original name below the image and the renovated name in the text box.

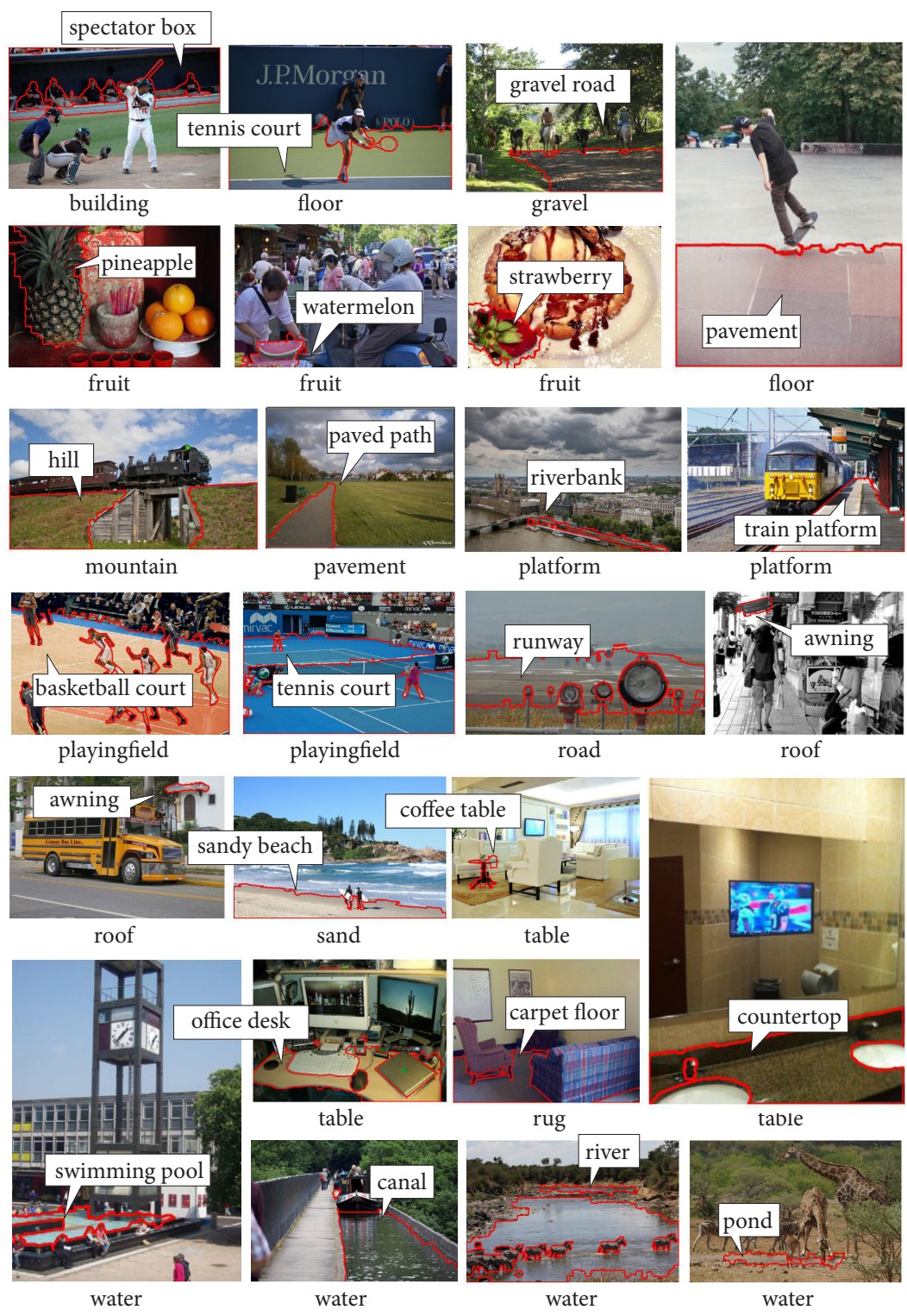

Figure E.2: **More examples of renovated names on segments** from the validation set of MS COCO. For each segment, we show the original name below the image and the renovated name in the text box.

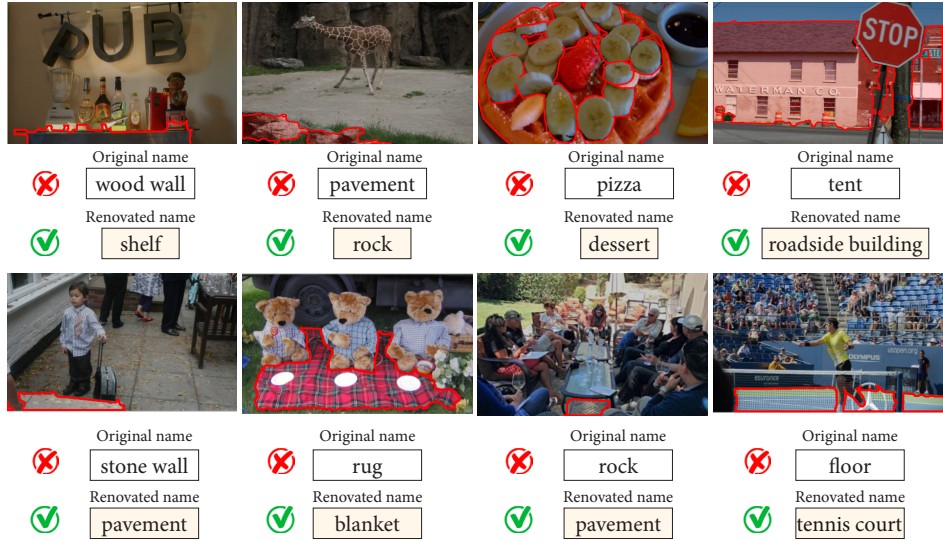

Figure E.3: **RENOVATE can find wrong annotations and suggest corrections.**

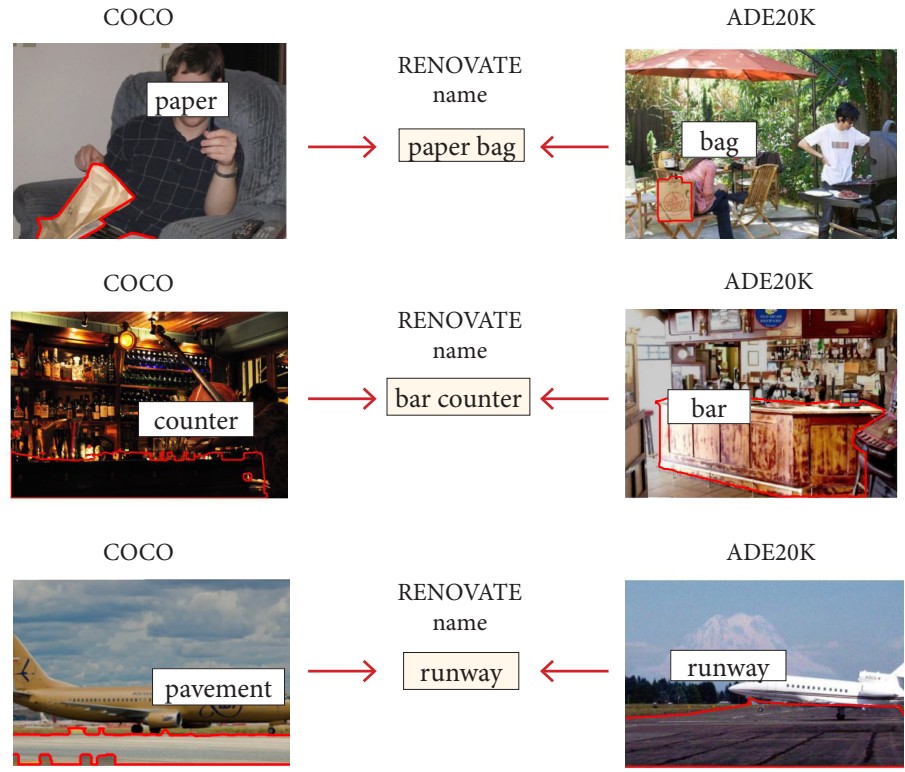

Figure E.4: **RENOVATE uncovers segments with shared semantic concepts across datasets.**

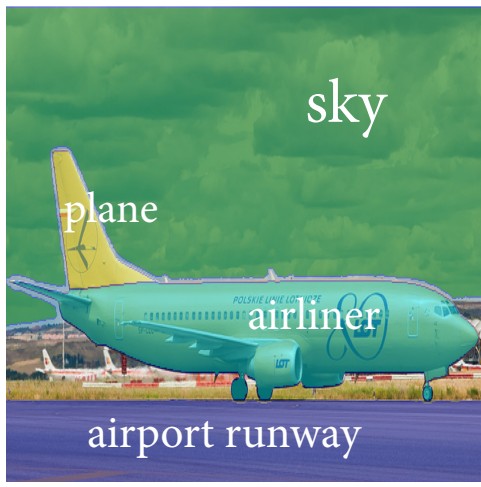

(a) Tags: air field, airliner, plane, airport runway, cloudy, floor, land, raceway, sky, tarmac, yellow

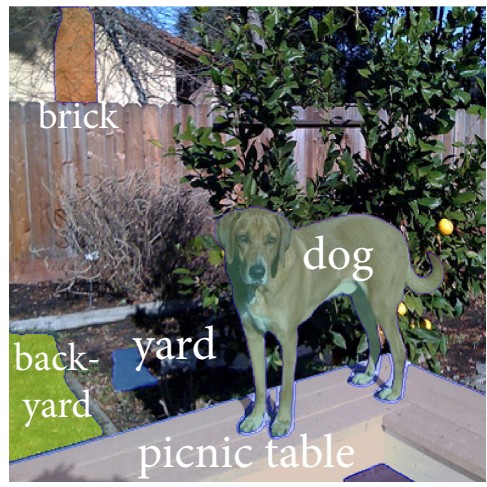

(b) Tags: backyard, ball, brick, dog, ledge, lemon, picnic table, ramp, stand, yard

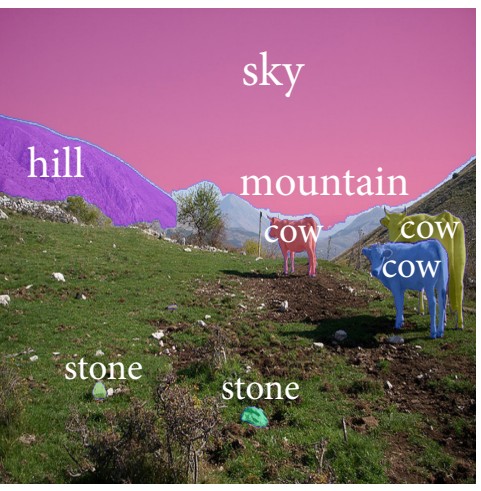

(c) Tags: animal, blanket, bull, cattle, trumpet, stone, cow, field, grass, grassy, green, herd, hill, hillside, lush, mountain, stand, white

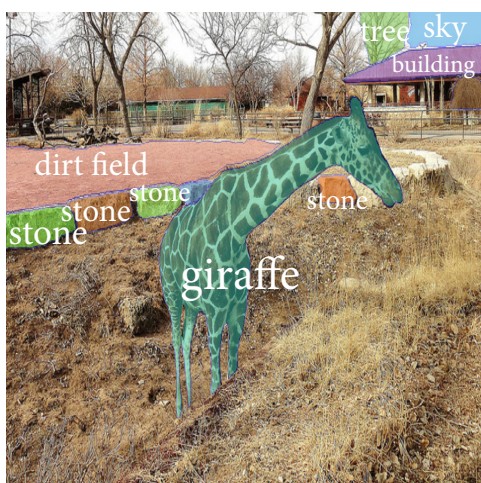

(d) Tags: area, dirt field, dry, enclosure, fence, field, floor, giraffe, grass, habitat, hay, building, sky, tree, zoo

Figure E.5: **RENOVATE can rename segments on generated masks and image tags.** We show the full list of the image-level RAM-generated tags as captions of each image and plot all SAM2-generated masks with their best-matching tags. Note some objects are not shown in the image as they are not segmented out by SAM2.

