# OpenReview forum: "Renovating Names in Open-Vocabulary Segmentation Benchmarks"
_NeurIPS.cc/2024/Conference — NeurIPS 2024 poster_

### Official Review · Reviewer_CT7Q · 2024-07-09

**Soundness:** 3
**Presentation:** 4
**Contribution:** 3
**Rating:** 6
**Confidence:** 5

**Summary:**

Open-vocabulary models use class names as text prompts for unseen categories’ generalization during training. In this paper, the issue of imprecise or even wrong class names from existing datasets is specially studied. One simple and general framework is proposed for automatic dataset renaming, that is, first narrow down the whole name search space to a curated list of candidate names, by using original class names with contextually relevant nouns from image captioning; next employ a specially trained renaming model to select the best-matching candidate name for every ground-truth segmentation mask. To demonstrate the effectiveness, two types of practical applications are explored, i.e. using renovated names to train open-vocabulary models, and applying renovated names to improve evaluation of open-vocabulary segmentation.

**Strengths:**

[+] Due to the fact that class names are often directly used as text prompts, they have a significant impact on the performance of open-vocabulary tasks. Therefore, focusing on the noisy issue of category names is on the right path.

[+] Leveraging foundation models to automate the renaming process indeed reduces the manual labor.

[+] The paper is easy to follow and understand, having clear logic.

[+] Some experiments are conducted to demonstrate that renovate names help to train models with stronger open-vocabulary capabilities and refine evaluation benchmarks.

**Weaknesses:**

[-] Noise type & noise rate. The authors partially visualize the noise issue of some class names, such as inaccurate or too general or lack contexts, which is worth encouraging. But, are there only these three types of noise in real-world scenarios? What are the noise rates for different datasets? And is the effectiveness of RENOVATE the same for different types/ratios of noises?
Thorough statistics are needed to clarify the performance bounds.

[-] How many ground-truth masks are needed when sorting candidate names using visual context? How to ensure consistency in visual alignment, as visual masks typically have significant intra-class variances (such as humans having different hairstyles)?

[-] To obtain candidate names, GPT-4 and image captioning are used. For GPT4, it is usually sensitive to prompts, how to ensure the quality of generating candidate names (Quantitative comparisons are needed)? Do you need additional labor to perform a double-check after combining results from GPT-4 and captioning?

[-] Since renovating names for open-vocabulary understanding seems to be one general idea, why only focus on segmentation tasks? Will the conclusion of this paper be exactly the same for classification or detection? For example, when using visual samples to match the best candidate, the intra-class visual variances of classification are definitely greater than those of segmentation.

**Questions:**

[-] In the related-work section, some citations are missing, which may mislead readers/reviewers. [1,2] have explored the text issue of class names in open-vocabulary segmentation, with some overlapping with this paper. The reviewer encourages including them and conducting detailed discussions and comparisons.

[1] AttrSeg: Open-Vocabulary Semantic Segmentation via Attribute Decomposition-Aggregation.      NeurIPS 2023

[2] Multi-Modal Prototypes for Open-Set Semantic Segmentation. Spring IJCV

**Limitations:**

As the authors said, RENOVATE could inadvertently propagate biases from the foundational models into the new names. To mitigate potential negative societal impacts, they advocate for verification of names in critical applications.

Also, the authors acknowledge that, the exploration remains incomplete, and will further make refinement to explore more model backbones and scale up to large-scale datasets.

---

> ### Author Rebuttal · Authors · 2024-08-06
>
> We thank the reviewer for acknowledging our paper to be “on the right path” and that our presentation to be “easy to follow” and have “clear logic”. We address the rest of the comments below.
>
> **1. Noise types and rates.**
>
> Besides the noise types we visualized in Figure 1, there are indeed many other noise types in the real world, such as “being too specific”, “being too ambiguous”, “being culturally/socially inappropriate”, and many more. However, since no groundtruth names are known, it is infeasible to determine exactly how many types of name mistakes are there. It is thus also impossible to determine the “noise rates” for each noise type.
> However, in the future, one can use human annotators or even VQA models to annotate a dataset for such name mistakes. That would certainly be very valuable for our understanding how noise in names affect the models.
>
> Nevertheless, we can still use human-verified RENOVATE names on the validation set to estimate the “noise rates” of different datasets. Specifically, we can define noise rate$=1- sim($original name, renovated name$)$ to assess how different the original names are from the RENOVATE names, and use the averaged score to assess the “noise rate” of a dataset. Under this definition, the noise degree of ADE20K and Cityscapes are 0.59 and 0.39, respectively. We can see that ADE20K exhibits a higher noise ratio compared to Cityscapes, potentially because ADE20K has more samples and classes and is likely to have more errors.
>
> **2. On the name selection process.**
>
> As written in L179-182, we perform name ranking and selection for each segment separately. Therefore, only one groundtruth mask is needed for each segment. Since we do per-segment renaming, there is no intra-class variance to deal with.
>
> **3-(a). Sensitivity to GPT4 prompts.**
>
> Our GPT prompts have the following components: (1) original name; (2) context names from image captioning; (3) suggestions on the two types of names (synonyms and short contexts) to focus, with examples; (4) instructions on how to make use of the original name, with examples.
>
> In Table B.1, we performed an ablation study on the different choices of the context names and we can see that better context names result in significant gains in the renaming performance. We further ablate components (3) and (4) by removing them from the GPT-4 prompt and report the results in the following table. From our empirical studies, we find that GPT-4 is typically robust to the specific wording of the four components and that removing any key component (esp (2) and (3)) can significantly affect the results.
>
> ||PQ|AP|mIoU|
> |-|-|-|-|
> |full prompt |**27.9**|**17.9**|**37.1**|
> |- (2) |25.5|16.5|33.8|
> |- (3)|25.6|16.8|34.0|
> |- (4)|26.5|17.3|35.2|
>
>
> **3-(b). Manual check for GPT-4 results**
>
> Our pipeline does not require additional manual checks at the GPT-4 results as they already incorporate knowledge from our “context names” and the original name.
>
> **4. Why only focus on the segmentation task?**
>
> We chose the segmentation task as this is one of the most representative tasks for 2D recognition. Please note our segmentation task covers semantic, instance, and panoptic segmentation. Our method is readily applicable to detection (by replacing the mask by the bounding box) and classification (by replacing the mask by the whole image).
>
> Specifically, we conducted an additional study on an open-vocabulary object detection model YOLO-World [1] and by replacing the original names with RENOVATE names for fine-tuning, we improved the performance on ADE20K from an AP of **18.1%** to **21.2%**, which is equivalent to 17.1% relative improvement. We will add the results to the paper.
>
>
> **5. More related works**
>
> We thank the reviewer for pointing out more related works. Both works mentioned by the reviewer propose to decompose the class names into a set of attributes or prototypes and are indeed also addressing the problem that class names are typically not descriptive enough for open-vocabulary segmentation tasks.
>
> However, note that our work addresses the problem of class names in a different way – our RENOVATE framework is able to deal with various kinds of problems with the original names and directly improve the name quality for the segmentation task.
> We will discuss these related works in our final camera-ready version.
>
> [1] Cheng, et al. "Yolo-world: Real-time open-vocabulary object detection." CVPR 2024.

---

> > ### Comment · Reviewer_CT7Q · 2024-08-11
> >
> > The reviewer thanks for the efforts made during rebuttal, and most of my issues has been solved.

---

> ### Author Response · Authors · 2024-08-11
> **Thanks for the response**
>
> We thank the reviewer for the response and we are pleased to see that our rebuttal has addressed the reviewer's concerns.
>
> If any remaining concerns hold the reviewer’s opinion on the current borderline recommendation, we would be happy to provide further clarification and discussion. Otherwise, we invite the reviewer to kindly raise the recommendation rating.
>
> Update: We have noticed the recommendation score raising and would like to thank the reviewer again for recognizing the value of this work.

---

### Official Review · Reviewer_aQgc · 2024-07-09

**Soundness:** 3
**Presentation:** 3
**Contribution:** 2
**Rating:** 6
**Confidence:** 4

**Summary:**

The paper explores modifying the class names associated with mask annotations in the segmentation datasets in the context of open-vocabulary segmentation. A method, RENOVATE, is described to perform such renaming in an automated pipeline. RENOVATE leverages an image captioning model to generate contextual words for each class. GPT-4 is prompted by class name and the contextual world to generate a pool of candidates. A segmentation model is trained to select among the candidate names using the mIoU as a proxy metric. The refined class names are then used to train open-vocabulary models, which show improvement in performance.

**Strengths:**

1) The paper explores an interesting angle of open-vocabulary segmentation: improving the names assigned to each segment. This is a novel study, highlighting the value of precise and varied captions for downstream training.
2) The writing of the paper is engaging. Particularly, the ample use of examples helps illustrate key points around the proper use of terms.
3) The paper conducts a very small scale study (n=20) human study showing a preference for new names.
4) The conducted ablation study verifies some components of the pipeline.

**Weaknesses:**

1) The section on the training of the ranking model to select among the candidate names could be clearer and more streamlined. It is not particularly clear whether a model is trained for each category or if a single model is trained for all categories.
   - If it is the former, then there are certainly some computational concerns and limitations that the method imposes. It would be useful to discuss the computation cost wrt to the number of classes in the data.
   - If it is the latter, then the ability to use or re-use this trained model for the downstream task of OV segmentation could be considered. It seems that the effect of gt-mask attn bias is limited (B.2) as randomisation performs better. Also, reporting the performance of this model on the downstream task is of interest.
2) In the same section, the design and presentation of a specialised architecture for the renaming could be better motivated. Since a proxy of a segmentation task is employed, is the new architecture even necessary? Could existing model architecture perform this task? Why do existing OV segmentation model pipelines not meet the requirements of the proposal's renaming stage?

3) The evaluation of the study for the improved performance of the segmentation models is limited -- only a single architecture is explored. Other architectures would help rule out particular drawbacks of the FC-CLIP being addressed with renaming and would provide more evidence to the broader conclusion about renaming helping model training (L310).

4) While the new names might help identify particular problems in models (L297), the reasoning behind renovating names in evaluation is rather backwards. While some misclassifications might be due to the similarity in labels, such as those identified as "benign" by the authors, they are still misclassifications as defined by the task. This then creates a problem in downstream use. For example, two masses in an X-ray might appear very similar due to both being a collection of cells, but their placement and small characteristic outline patterns will determine the difference between benign and dangerous growths. Thus, one should seek to highlight key errors in the evaluation instead, even if the labels are close. It seems that a different phrasing of the conclusions (L296): the learned semantics are not sufficient to differentiate between semantically adjacent classes.

5) [Minor] Finally, the proposed pipeline itself might be less applicable outside of the academic where established datasets with limited label sets are used. Would gathering a dataset with varied captions from the onset make the RENOVATE method redundant?

**Questions:**

- It would be good if the issue surrounding the training of the renaming model (W1) could be clarified in the rebuttal.
 - Additionally, providing some commentary on the weaknesses (W2) and (W4) would help improve the interpretation of the paper.
 - Furthermore, it would be good to discuss future applicability of the pipeline, or whether the better path is to just collect more precise captions for training data.

Overall, the paper is well-written and presents some insights into the treatment of data for open-vocabulary segmentation. There are some limitations in the evaluation. I currently rate the paper as BA prior to rebuttal.

**Limitations:**

The paper touches on the potential to propagate existing biases in the foundational models it employs, which is a big issue, and it is important that it is mentioned.

I would recommend conducting an ethics review or obtaining IRB approval for the human study, even if it is not strictly necessary according to the respective institutional guidelines.

---

> ### Author Rebuttal · Authors · 2024-08-06
>
> We thank the reviewer for acknowledging the novelty of our paper and our efforts in making our writing “engaging”. We address additional comments as the following:
>
> **1. On the renaming model and downstream segmentation task.**
>
> Thanks for the suggestion, we will work on making this part more streamlined and clear. Now to answer the specific questions:
>
> First, we train the renaming model for all categories of a dataset.
>
> Second, it is an interesting idea to use the renaming model for downstream segmentation. To do so, we need to use it without gt-mask attn biases and use names from all the classes as the text queries. Note that the “rand gt mask” in Table B.2 still uses gt masks – it only randomizes in the intermediate layers of the transformer decoder, but always uses gt masks at the initial layer. Using a renaming model without gt masks would compromise the performance significantly and make the model unable to distinguish different instances from the same class, as reflected in its segmentation performance on ADE20K: an mIoU of **33.2%** (semantic segmentation) and AP of only **12.9%** (instance segmentation). Similarly, OVSeg models cannot perform as well as our renaming models in the renaming task as they do not make full use of the available knowledge for renaming (see discussion in **2.**).
>
> While the renaming model may not be useful for downstream segmentation, it can be used for other tasks such as **generating mask-name annotations for image datasets** by matching caption nouns or tags with mask proposals, making it potentially useful to improve many large-scale datasets. See the PDF for an example.
>
> **2. The necessity of an architecture for renaming.**
>
> Compared to OVSeg, the renaming task has more knowledge of the inputs (gt masks and gt classes) and only focuses on finding optimal names within candidate names of each class for each visual segment. Thus, when designing renaming model architecture, it is a good idea to make use of these different task requirements to ensure high-quality renaming results.
>
> In fact, our renaming architecture is nearly identical to Mask2Former except that we use text embeddings as input queries and gt masks (with randomization in intermediate layers) as attention biases for cross-attention computation. As shown in Table B.2, both modifications are essential to improve the renaming performance (e.g., from an mIoU of 27.1% to 37.1%). We further reported the renaming performance when using OVSeg architecture FC-CLIP under the same setup as in Table B.2:
>
> ||PQ|AP|mIoU|
> |-|-|-|-|
> |FC-CLIP (for renaming)|19.6|14.3|25.3|
> |Our renaming model |**27.9**|**17.9**|**37.1**|
>
> We can see that our renaming architecture is indeed much superior at the renaming task.
>
> **3. Other architectures for training with the RENOVATE names.**
>
> We thank the suggestion for extending the current training experiments to include more architectures.
>
> We further experiment with a very different architecture, YOLO-World[1], which does not rely on a CLIP visual backbone. To compare the training name quality on COCO, we fine-tune YOLO-World-M on the COCO object detection dataset and evaluate OV object detection on ADE20K. By replacing the original names with RENOVATE names, we improve the AP from **18.1%** to **21.2%**, which is equivalent to 17.1% relative improvement. This further demonstrates the general applicability of RENOVATE names to help model training.
>
> **4. On the evaluation with renovated names.**
>
> The reviewer raised an interesting misclassification case where the visually close misclassifications are of high interest and suggested that “one should seek to highlight key errors in the evaluation”. We totally agree with this.
>
> We want to first point out that the “key errors” are highly domain-dependent. For example, in autonomous driving, identifying a “car” as a “truck” is much less dangerous than identifying it as a “road” for an autonomous driving car. In this case, the key error is not “car/truck” but “car/road” misclassification. In this domain, classification mistakes with high semantic differences (e.g., “car/road” mistake) are often thought of as more “wrong” as it is consistent with our intuition of a classification mistake.
>
> In the X-ray case, we can use mistakes in “placement and small characteristic outline patterns” as the key errors instead of “visual similarity” and design the evaluation metric accordingly.
>
> Finally, regardless of how to design the evaluation metric, renovated names make it possible to conduct fine-grained analysis on mistakes with different semantic distances. Without our renovated names, we can only see the coarse misclassifications without a fine-grained understanding of the model.
>
> We will add this discussion of the evaluation metric in our revision.
>
> **5. Further applicability of the renaming pipeline with rich captions.**
>
> We want to first point out the fact that collecting a high-quality multi-caption dataset is very expensive — current caption data collection is mainly from web scraping which is of low quality, often misses many objects in the scene, and doesn’t consist of multiple/rich annotations. Curating these datasets already requires a lot of effort and costs[2,3].
>
> Compared to this, it is very economically efficient to curate a dataset with relatively coarse annotations and use our RENOVATE to **improve the quality of annotations**. From this perspective, RENOVATE is indeed very useful and meaningful to improve current large-scale datasets with coarse web-scraped annotations. In the PDF, we showed an example where **the renaming model is applied to generated masks and tags and produces good mask-name matches**. This shows that RENOVATE has broad future applicability.
>
>
> [1] Cheng, et al. "Yolo-world: Real-time open-vocabulary object detection." CVPR 2024.
>
> [2] Gadre, et al. "Datacomp: In search of the next generation of multimodal datasets." NeurIPS 2024.
>
> [3] Fang, et al. "Data filtering networks." arXiv preprint (2023).

---

> > ### Comment · Reviewer_aQgc · 2024-08-10
> > **Thank you for the rebutall**
> >
> > I wanted to acknowledge and thank the authors for their detailed responses. I believe the majority of my concerns have been addressed, and I support the proposed changes. I will update my recommendation accordingly.

---

> > > ### Author Response · Authors · 2024-08-11
> > > **Thanks for the Response**
> > >
> > > Thank you for taking the time to review our response. We’re pleased that our rebuttal has addressed your concerns and we appreciate the recommendation score increase. We will make sure to incorporate the reviewer's suggestions into our revised paper.

---

### Official Review · Reviewer_fwux · 2024-07-12

**Soundness:** 3
**Presentation:** 2
**Contribution:** 3
**Rating:** 6
**Confidence:** 4

**Summary:**

The paper considers the problem of open-vocabulary segmentation. The task requires segmentation models to recognize categories outside of the training taxonomy. This usually means learning a joint image-text feature space and classification via similarity matching between class text embeddings and segmentation mask embeddings. The paper aims to improve the suboptimal text descriptions of the dataset classes, which are often too general, ambigous and context dependant. The proposed method describes a strategy for automated renaming of the assigned class descriptions at the segment level. This means that each ground truth segment gets assigned with one or more text descriptions that describe that particular segment more specifically. Training with such enriched text descriptions improves the model performance on both the source and target datasets. The experimental study follows the usual setup where COCO is used as source dataset, and ADE20k and Cityscapes as target datasets.

**Strengths:**

The paper considers an important and underexplored problem in open-vocabulary segmentation. To the best of my knowledge, OpenSeg [5] is the only prior work that also considers the problem of suboptimal class descriptions and corresponding renaming solutions. The paper provides a proper comparison and discussion of the differences.

The proposed renaming method is fully automated, which represents a significant step forward w.r.t. to the related work that requires manual inspection. The described approach is technically sound and properly explained.

The proposed method seems dataset and model agnostic. This means that it should easily generalize to other datasets and would be beneficial for training of different open-vocabulary segmentation models. The authors promise to share the code and the renamings for popular datasets which would have positive affect on future research in open-vocabulary segmentation.

The results reveal that training with renovated vocabulary leads to increased segmentation performance across different metrics and datasets. Furthermore, including the renovated vocabulary only in the evaluation phase leads to increased performance of the open-vocabulary models pretrained with regular datasets.

**Weaknesses:**

Some parts of the paper remain unclear.

For example, I am confused about the batch collation for training the renaming model. Figure 2 suggests that the transformer decoder in a single forward pass receives candidate names from a single ground truth class. Does this mean that for a single image the number of forward passes through the transformer decoder is equal to the number of ground truth segments? Would you say then that a single training example is consisted of an image, a single ground truth segment and multiple candidate names? Is then a batch size equal to the total number of segments in the sampled images? Does the batch size 16 from line 239 refers to the number of images or number of segments? Does this type of training affects the memory requirements significantly? Are there any redundancy in terms of the tensor representations that have to be kept in the memory in order to enable parallel training on multiple segments? I suggest authors to clarify these questions.

I also dont understand the details of using renovated names during the evaluation. Does this mean that all the candidate names are just appended to other naming variants of that particular class? It seems to me that we do not need to train the renaming model if we would like to use the renovated names just for the evaluation. Is this correct? How many naming variants per class do you get on average when you concatenate the proposals from the original set, the OpenSeg and the renovated names? Does this affect the inference speed significantly? If you also use the usual templates (a picture of *class name*, an image of *class name*, etc.) it seems to me that there might be a lot of different textual representations for a single class. How does this inference really work?

The renaming model should be compared to some stronger baseline which also has the ability of candidate selection. For example, you could assign each segment with a candidate name that achieves highest similarity with the segment representation based on the CLIP embeddings. The segment representation could be average-pooled feature representation from the CLIP visual encoder. I question the necessity of training the mask prediction module just for renovating the class name associated with some segment, if we already know the corresponding ground truth mask. Could you please comment on this issue and maybe discuss some possible alternatives?

In Table 3. the results obtained with OpenSeg names should correspond to the original FCClip results. However, in some cases the reported performance is lower than the FCClip performance reported in the official repository. Why is this the case?

As shown in Table 3., training with renovated names leads to marginal gain in some cases (e.g. PQ on Cityscapes).

I am not sure if the x-axis on figure 5 shows the number of images available in the training dataset, or the number of iterations trained on the full dataset? If these experiments consider subsampled training subsets, you should explain how the training images were selected.

I am a bit surprised about the effectiveness of the negative sampling. What would be the effect of including additional renovated names of the ground truth class as negatives? This reminds of hard example mining, as it would force the model to differentiate between similar semantic concepts.

**Questions:**

Please consider the questions from the weaknesses section that would clarify training of the renaming model and the evaluation procedure with the renovated names. Also, please discuss the possibility of evaluating the renaming model with another baseline with candidate name selection ability. Please find the suggestion in the weaknesses section, which considers a baseline based on pooling of CLIP features.

**Limitations:**

The authors discussed the limitations in the section 6. The main concern is regarding the bias transfer from the foundation models.

---

> ### Author Rebuttal · Authors · 2024-08-06
>
> We thank the reviewer for pointing out that we addressed an “important but underexplored” problem and that our approach is “technically sound”, “properly explained”, and “would have positive impact on future research”. We address the remaining comments below.
>
> **1-(a). Batch collation to train the renaming model**
>
> Our renaming model processes all images in a batch (16 images) in a single forward pass – all the text queries from the gt segments are processed in parallel. Specifically, each input query is to assess the alignment between one gt mask and one of its candidate names — the input query itself is initialized with the text embedding of the candidate name, and it is refined by the cross-attention layers that use the gt mask as attention biases.
>
> Since the model can process multiple queries in a single pass (like Mask2Former), we can process all such mask-name pairs in one batch in a single pass. For images where the number of pairs is less than the number of queries, we simply pad with empty queries for collation.
>
> We will make the batch collation more clear in the paper.
>
> **1-(b). Memory usage**
>
> Our model **consumes similar memory** to a standard Mask2Former model. For the memory consumption, the key factor is the number of queries. In Mask2Former, this number is 250. In our renaming model, it usually varies between 100 to 350, depending on the number of candidate names and segments in images in a batch.
>
> **2-(a). Name merging for evaluation in Table 3**
>
> For results in Table 3, when evaluating, we append the **“renovated names”** instead of **“candidate names”** to the other naming variants. After merging the names, the number of names per class is 6.44, 4.98, and 6.89 for COCO, ADE20K, and Cityscapes, respectively. This has barely any influence on inference time (0.3sec/img) since the **inference cost is mostly in processing the image and query features** instead of the final cosine similarity computation. One can even choose to use more test names (e.g. 10 names per class) depending on the use case and will not incur any noticeable extra cost of inference time.
>
> **2-(b). The usage prompt template in inference**
>
> For each prompt name, we first use ViLD templates to create 63 different sentences and extract their CLIP text embeddings. We then average the text embeddings after normalization and use this **single embedding** as the text embedding for the corresponding name. This is also the common practice in other OVSeg works [1,2].
>
> **2-(c). The need of renaming model for the evaluation**
>
> Since we need “renovated names” for evaluation, we need to train the renaming model to obtain them. Evaluating using candidate names would introduce unnecessary complications in the evaluation as they are quite noisy (as indicated by the inferior performance when training with candidate names in Table 3).
>
> **3-(a). Baselines for the renaming model**
>
> We thank the reviewer for the suggestion of using average-pooled CLIP features as a baseline. In prior works, this baseline is known to be inferior to tailored OVSeg models[1]. In [1], it’s shown that with gt masks as region proposals, average-pooled CLIP features only reach an mIoU of **20.1%** on ADE20K, being significantly lower than OVSeg models (at least over 30%).
>
> Our baseline for the renaming model is the FC-CLIP with gt attn biases (row 1 in Table B.2), which is much stronger than the average-pooled CLIP features. Our model significantly improves upon this baseline (from **27.0%** mIoU to **37.1%** mIoU) by incorporating our designs of “rand gt masks” and “text queries”.
>
> **3-(b). Necessity of training the mask prediction module.**
>
> Relatedly, since CLIP features are not good enough, we need to improve them. The pixel decoder in our renaming model is responsible for improving the CLIP features to a per-pixel level. The mask prediction module is irreplaceable since it’s the only source of supervision signals for these per-pixel features.
>
> We also note that even if we use gt masks as attn biases, the mask prediction is still non-trivial. As we write in L160-164, the gt masks are only guiding the cross-attention to focus on the segment regions.
>
>
> **3-(c). Alternatives of the mask prediction module**
>
> As an alternative to our mask prediction module, one may design different proxy losses to train the renaming model. For example, a region-level contrastive loss that forces matched mask-name embedding pairs to be closer. As our focus of this paper is to demonstrate the importance of names and renaming, we leave such exploration to future work.
>
> **4. Table 3 OpenSeg results are lower than the original FCCLIP results.**
>
> This is because the performance of OV Seg model depends on the test name set. In Table 3, our test names merge all three name sets. This is for a fair comparison of models trained with different names. We will update the paper accordingly to make this clear.
>
> **5. Figure 5 setups.**
>
> The x-axis in Figure 5 shows the number of images available in the training set. The training images are randomly selected from the whole training set.
>
> **6. Negative sampling variants**
>
> Negative sampling is highly effective to train with a large vocabulary in NLP. Recently, some OVSeg models also have adopted a similar approach to ours [2].
>
> While negative names from other classes are guaranteed “true negatives”, same-class renovated names can be complicated. For example, for one segment, “man” might be the most precise, but “person” is also not wrong. It is thus not ideal to penalize “person” in the same way as names from other classes (e.g., “car”) during training. To remedy this, one may add an extra step to first select “true negatives” among each class. This can indeed be an interesting future extension of our current sampling approach.
>
> [1] Liang, et al. "Open-vocabulary semantic segmentation with mask-adapted clip." CVPR 2023.
>
> [2] Cheng, et al. "Yolo-world: Real-time open-vocabulary object detection." CVPR 2024.

---

> > ### Comment · Reviewer_fwux · 2024-08-12
> >
> > I would like to thank the authors for the rebuttal. Most of my concerns were addressed. Thus, I will upgrade my decision to weak accept.

---

> > > ### Author Response · Authors · 2024-08-12
> > > **Thanks for your response and score upgrade**
> > >
> > > Thank you for taking the time to review our response. We’re pleased that our rebuttal has addressed your concerns and we sincerely appreciate the recommendation score upgrade. We will carefully consider and incorporate the key points you've raised as we work on the revised paper.

---

### Author Rebuttal · Authors · 2024-08-06

We’d like to thank all reviewers again for their valuable feedback.

We are pleased to see that reviewers found that our work is **novel** (Reviewer aQgc) and **technically sound** (Reviewer fwux), **considers an important and underexplored problem** (Reviewer fwux, Reviewer CT7Q) and would **positively affect on future research** (Reviewer fwux). We are equally glad that the reviewers found the paper writing **engaging** and **easy to follow and understand** (Reviewer aQgc, Reviewer CT7Q).

We’d like to highlight a few points in our responses to the reviewers:

1. We added an experiment using YOLO-World [1] to train with renovated names and reported performance gains in open-vocabulary object detection on ADE20K from an AP of **18.1%** to **22.2%**, an equivalent of 17.1% relative improvement. This demonstrates the benefit of RENOVATE names in training different model architectures and different tasks (open-vocabulary object detection). See responses to **Reviewer aQgc** and **Reviewer CT7Q**.

2. We added a figure in the PDF that showcased how the renaming model can be used **without gt mask annotations** but **with SAM2-generated masks [2] and RAM-generated image tags [3]**. This demonstrates another possible application of the renaming model and shows that our renaming approach can be further generalized to datasets without mask annotations. See response to **Reviewer aQgc**.

3. We added ablation experiments on different components of the GPT-4 prompts for a better understanding of how GPT-4 prompts influence the renaming performance. See response to **Reviewer CT7Q**.

4. We added an experiment using FC-CLIP (without gt-mask attn bias during training) for renaming. Its significantly inferior performance to our renaming model suggested a need for our designated renaming model. See response to **Reviewer aQgc**.

5. We also added more implementation details on the training of the renaming model, the name selection process of the renaming model, and the evaluation of the models trained with renovated names. We hope our response clarifies the questions from the **Reviewer fwux** and **Reviewer CT7Q**.

We hope our replies to all the reviewers satisfactorily address their questions and comments and we warmly welcome further questions during the discussion phase.

[1] Cheng, et al. "Yolo-world: Real-time open-vocabulary object detection." CVPR 2024.

[2] Ravi, Nikhila, et al. "SAM 2: Segment Anything in Images and Videos." arXiv preprint arXiv:2408.00714 (2024).

[3] Zhang, Youcai, et al. "Recognize anything: A strong image tagging model." CVPR 2024.

---

> ### Author Response · Authors · 2024-08-14
> **Thanks again to all the reviewers**
>
> As the discussion period concludes, we would like to express our sincere gratitude to all the reviewers for their valuable feedback and suggestions. We are pleased to see that all the reviewers acknowledged our rebuttal effectively addressed their concerns and subsequently raised their recommendation ratings, recognizing the value of our paper. We will carefully consider all the feedback and incorporate it into our revised paper.

---

### Decision · Program_Chairs · 2024-09-25

**Decision:**

Accept (poster)

**Comment:**

In this paper, the authors presented a renaming model for open-vocabulary segmentation. Due to the precision concern of the class names in existing datasets, the authors were motivated to enhance their quality for visual segments. Experiments show that the renovated names help the training of stronger models for open-vocabulary segmentation. The main contribution of this paper is the study of the naming precision of existing datasets for the open-vocabulary segmentation task, which could benefit follow-up research in this field.

The paper received comments from three expert reviewers, and all of them acknowledged the contributions of the important problem that this paper looked into, and the corresponding renaming approach. However, there were some concerns around the unclear design of the approach and motivation, missing comparison to alternative methods/baselines, and experimental evaluations, leading to a mixed recommendation of both positive and negative. After the rebuttal together with a few rounds of discussions among the authors, reviewers and AC, the above major concerns were addressed and the reviewers agreed on the contributions of this paper and its potential impact on the field.

Considering the above-mentioned points, the discussions, and the potential interests of the paper may have on the NeurIPS audience, the AC recommends Accept, while reminding the authors to include the additionally provided clarifications and revisions during the rebuttal to the final version of the paper.